# Divide-and-Conquer Meets Consensus:
# Unleashing the Power of Functions in Code Generation

**Jingchang Chen**[*]
Harbin Institute of Technology
jcchen@ir.hit.edu.cn

**Hongxuan Tang**[*]
Harbin Institute of Technology
jeffswt@outlook.com

**Zheng Chu**
Harbin Institute of Technology
zchu@ir.hit.edu.cn

**Qianglong Chen**[†]
Zhejiang University
chenqianglong.ai@gmail.com

**Zekun Wang**
Harbin Institute of Technology
zkwang@ir.hit.edu.cn

**Ming Liu**[†]
Harbin Institute of Technology
mliu@ir.hit.edu.cn

**Bing Qin**
Harbin Institute of Technology
qbin@ir.hit.edu.cn

## Abstract

Despite recent progress made by large language models in code generation, they still struggle with programs that meet complex requirements. Recent work utilizes plan-and-solve decomposition to decrease the complexity and leverage self-tests to refine the generated program. Yet, planning deep-inside requirements in advance can be challenging, and the tests need to be accurate to accomplish self-improvement. To this end, we propose FUNCODER, a code generation framework incorporating the divide-and-conquer strategy with functional consensus. Specifically, FUNCODER recursively branches off sub-functions as smaller goals during code generation, represented by a tree hierarchy. These sub-functions are then composited to attain more complex objectives. Additionally, we designate functions via a consensus formed by identifying similarities in program behavior, mitigating error propagation. FUNCODER outperforms state-of-the-art methods by +9.8% on average in HumanEval, MBPP, xCodeEval and MATH with GPT-3.5 and GPT-4. Moreover, our method demonstrates superiority on smaller models: With FUNCODER, StableCode$_{3b}$ surpasses GPT-3.5 by +18.6% and achieves 97.7% of GPT-4's performance on HumanEval. Further analysis reveals that our proposed dynamic function decomposition is capable of handling complex requirements, and the functional consensus prevails over self-testing in correctness evaluation.

## 1 Introduction

Over the past few years, large language models have been observed to attain significant advancements in coding capabilities (OpenAI, 2023; Touvron et al., 2023). Meanwhile, models designed specifically for coding tasks have also been introduced (Rozière et al., 2023; Lozhkov et al., 2024; Pinnaparaju et al., 2024). Although LLMs can proficiently generate simple code snippets, they suffer from a decline in performance as code requirements become complicated.

Numerous efforts have been made to tackle this complexity. The two-stage methods (Jiang et al., 2023; Zelikman et al., 2023) employ the plan-and-solve strategy, which first generates a draft outline

---

[*]Equal contribution.
[†]Corresponding Authors: Ming Liu, Qianglong Chen.

38th Conference on Neural Information Processing Systems (NeurIPS 2024).

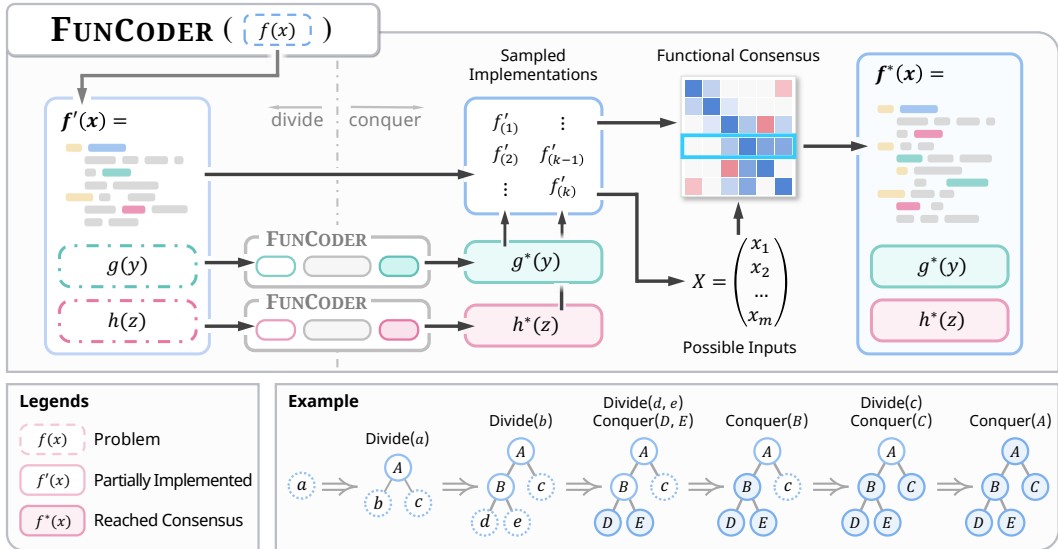

Figure 1: A flowgraph illustrates FUNCODER. FUNCODER branches off new functions to have sub-goals tackled iteratively (left), re-composites sub-functions, and selects the best using functional consensus (right). Bottom-right figure shows how FUNCODER writes functions at hierarchy-level.

for the complex task and uses it as guidance for implementing the code in the second stage. Multi-agent development frameworks (Hong et al., 2024; Qian et al., 2023) mimic real-world software development workflows, assign different roles to LLMs and collaborate to solve a complex goal. Self-improvement (Shinn et al., 2023; Chen et al., 2024), on the other hand, refines the program in accordance with execution feedback from self-generated unit tests.

Despite fruitful efforts made by the previous methods in dealing with complex problems, certain challenges still remain unsolved: (1) Two-stage approaches need to design a complete plan at the beginning and lack the ability to adjust the top-level design during implementation, leading to sub-optimal decomposition. (2) Multi-agent collaboration frameworks are cumbersome and rely heavily on LLM capabilities, making them difficult to generalize to smaller open-source models. (3) Code refinement through self-tests depends on the correctness of generated unit-tests. Our preliminary study (§3.1.3) finds that models generate unreliable self-tests in abundance. These incorrect tests may mislead self-improvement and, at worse, exacerbate program errors.

To address these issues, we propose FUNCODER, a code generation framework utilizing a divide-and-conquer strategy and a novel functional consensus mechanism on functions to decompose complex problems. Starting from the main problem, FUNCODER introduces new functions to cope with certain sub-problems. The new functions will be decomposed recursively, eventually forming a tree of functions. FUNCODER then combines functions bottom-up to achieve increasingly complicated objectives. By dividing-and-conquering tasks into simpler sub-functions, complexity can be gradually reduced. However, errors in sub-functions may propagate to the whole program, thereby damaging overall reliability. We propose functional consensus that samples multiple functions and selects the one demonstrating consensus, measured by the aggregated similarity among candidates. By reaching a consensus, we reduce the discrepancies in code behavior and thus alleviate cascading errors.

We conduct extensive experiments on code generation benchmarks (Chen et al., 2021; Austin et al., 2021; Khan et al., 2023) with GPT-3.5 (Ouyang et al., 2022) and GPT-4 (OpenAI, 2023), outperforming state-of-the-art methods by $+\mathbf{9.8\%}$ on average. Experiments are further carried out on the mathematical competition benchmark, MATH (Hendrycks et al., 2021b), achieving a $+\mathbf{6.0}$ improvement with GPT-4, indicating that FUNCODER can also generalize to complex reasoning. Our method is observed to be equally effective on open-source models (Meta AI, 2024; Mistral AI, 2024; Pinnaparaju et al., 2024; Rozière et al., 2023; Lozhkov et al., 2024), with an average gain over baseline of $+\mathbf{31.5\%}$ on HumanEval and $+\mathbf{47.7\%}$ on MATH. Additional analysis also shows the advantage of both divide-and-conquer and functional consensus. Our code is made openly available at https://github.com/cometeme/funcoder.

**Algorithm 1** FUNCODER procedure

**Require:** Entry func, $f_{\text{root}} = \{h_{\text{root}}, d_{\text{root}}, \phi\}$
**Require:** Large language model, LLM
1: **function** FUNCODER($f_{\text{cur}}$)
2:     — *Divide* —
3:     $f'_{\text{cur}}, \{f_i\} \leftarrow$ EXTRACT(LLM($f_{\text{cur}}$))
4:     **for** $f_i \in \{f_i\}$ **do**
5:         **if** $b_i$ **is** NOTIMPLEMENTED **then**
6:             $f_i^* \leftarrow$ FUNCODER($f_i$) ▷ recursion
7:         **end if**
8:         ADDCHILD($f_{\text{cur}}, f_i^*$)
9:     **end for**
10:    — *Conquer* —
11:    $F_{\text{cur}} \leftarrow$ SAMPLE(LLM($f'_{\text{cur}}$, CHILD($f_{\text{cur}}$)))
12:    $f_{\text{cur}}^* \leftarrow$ FUNCONSENSUS($F_{\text{cur}}$)
13:    **return** $f_{\text{cur}}^*$
14: **end function**
15: **return** FUNCODER($f_{\text{root}}$)   ▷ starts from root

Figure 2: Left: Algorithm for FUNCODER, explained in detail in Appendix A.6. Right: Comparison between decomposition by planning and our approach. FUNCODER introduces new functions to describe sub-goals solely with code, achieving a more natural way of requirement decomposition.

## 2   FUNCODER: Divide-and-Conquer Meets Consensus

### 2.1   Divide-and-Conquer for Iterative Programming

A function is defined as a relation between a set of inputs and outputs where each input is assigned exactly one output (Halmos, 1998), denoted as $y = f(x)$. In computer programming, a function is identified by its header $h_f$ with its body $b_f$, and is commonly accompanied by a documentation $d_f$ to improve readability. Functions can be invoked from other procedures, allowing for the decomposition of large and complicated requirements into smaller structures that exhibit high comprehensibility and quality (Dahl et al., 1972). Generally, human programmers tend to decompose tasks into clearly defined sub-functions and then implement them recursively, making functions eligible for re-usage, taking advantage of the *divide-and-conquer* principle. Inspired by this, FUNCODER recursively *divides* the requirement and *conquers* functions to formulate a sophisticated solution, unleashing the potential of LLMs in code generation.

**Divide** is a top-down process that iteratively breaks down problems. Given a code generation problem, the process begins from the entry function $f_{\text{root}}$. We instruct the model to introduce new functions $f_i \in$ CHILD($f_{\text{cur}}$) that solve certain sub-goals while writing the current $f_{\text{cur}}$. To reduce the complexity involved in each generation, we only require the headers $h_{f_i}$ and documentation $d_{f_i}$ of new functions to be generated, while their implementations $b_{f_i}$ can be postponed. After completing the current function, the model starts to address those unimplemented sub-functions and complete $b_{f_i}$ into $f'_i$. This process stops when the model deems functions too simple to be further divided, finally forming a dependency tree $T =$ TREE($f_{\text{root}}$, CHILD($f_{\text{root}}$)). The *divide* process is similar to a search starting from the entry function, gradually involving new sub-functions while writing the current, and implementing them recursively. We guide the entire process through a depth-first search.

**Conquer** is a process of achieving complex objectives through aggregating smaller functions. We notice that child functions are not yet implemented during the top-down process of writing parent functions. As a result, these parent functions may not be able to effectively utilize the child functions, or misuse them at worst. FUNCODER deals with this issue by re-generating functions in inverse topological order on the dependency tree $T$ - starting from leaves, complex goals are handled by compositing solved children as $f_{\text{cur}}^* \leftarrow \mathcal{F}(f'_{\text{cur}}, \{f_1^*, f_2^*, \dots\}) \mid f_i^* \in$ CHILD($f_{\text{cur}}$).

*Divide* and *conquer* naturally achieve both decomposition and composition during code generation. Unlike two-stage and agent-based methods, our approach dynamically introduces new functions

along the process, making it less burdensome than producing a complete plan at the very beginning. Moreover, while planning or agents require chat capabilities, FUNCODER represents sub-tasks through functions (Figure 2), making it more applicable to specialized code generation models.

## 2.2 Functionality Similarity as a Consensus

The decomposition of complex tasks benefits from solving easier sub-goals, but might introduce the risks of cascading errors, which refers to errors in sub-functions that lead to errors in ancestral functions. To mitigate this, we introduce Functional Consensus which aims at reducing inconsistencies in program behavior. This is achieved by sampling multiple functions and selecting the one that exhibits consensus, as measured by the aggregated similarity of functionality between candidates, thus abating outlier functionalities.

**Functionality Similarity** A program specifies its functionality (or behavior) through the control flow and logic defined by its code semantics. However, comparing the functionalities between two programs based on their semantics is somewhat challenging. By decomposing the requirement into functions, FUNCODER is able to view the function behavior as a black box that maps arguments into return values. Considering two functions $f$ and $g$ with the same input domain $D(f) = D(g)$, we define the similarity between them $sim(f, g)$ as the identicalness of outputs when given the same input values.

$$sim(f, g) = \int_{x \in D(f)} \frac{\mathbb{1}\left[f(x) = g(x)\right]}{|D(f)|} \approx \sum_{x \in X | X \sim D(f)} \frac{\mathbb{1}\left[f(x) = g(x)\right]}{|X|} \tag{1}$$

The similarity becomes 1 if and only if two functions output consistent values for all inputs: $\forall x \in D(f) : f(x) = g(x) \Leftrightarrow sim(f, g) = 1$. We notice that the input domain $D(f)$ is unbounded in most cases, making its measurement barely feasible in practice. Thus, we approximate it by sampling a subset of possible inputs $X \sim D(f)$ with an LLM.

**Consensus** is reached by selecting the candidate $f^*$ holding maximal similarity with others after sampling multiple function implementations $F = \{f_{(i)}\}$ for the same requirements.

$$f^* = \text{FUNCONSENSUS}(F) = \underset{f_{(i)} \in F}{\arg\max} \sum_{f_{(j)} \in F \setminus \{f_{(i)}\}} sim(f_{(i)}, f_{(j)}) \tag{2}$$

By introducing functional consensus, FUNCODER produces functions that are more consistent and common in functionality, while omitting abnormal samples. The process is applied to not just the final program, but also to every sub-tree during the bottom-up *conquering* stage, resulting in step-by-step, thorough verification from the most fundamental functions all the way up to the whole program.

## 2.3 FUNCODER is a Function Coder

We design FUNCODER as a procedure that takes a problem in the form of a function signature $f(x)$, and produces a final solution $f^*(x)$, as exemplified in Figure 1. Given a problem $f(x)$, FUNCODER partially implements the function as $f'(x)$ referring to unimplemented sub-functions $g(y)$ and $h(z)$. These sub-functions are then fed into FUNCODER to be recursively coped with. We then sample $k$ implementations $f'_{(i)}(x)$ based on solved children $g^*(y)$ and $h^*(z)$. Functional consensus is calculated by evaluating candidates on possible inputs. The function sharing maximal behavioral similarity is combined with solved children to formulate the final solution.

## 3 Experiments

We conduct experiments on competition-level code generation and mathematical reasoning benchmarks with state-of-the-art LLMs, which are covered in section §3.1 and §3.2, respectively. In addition to GPT models (Ouyang et al., 2022; OpenAI, 2023), we also conduct experiments with community models like Llama3$_{8b}$ (Meta AI, 2024), StableCode$_{3b}$ (Pinnaparaju et al., 2024), and CodeLlama$_{34b}$ (Rozière et al., 2023). We use the *instruct* variant of these models and inference on a single A100-80G under BF16 precision with vLLM (Kwon et al., 2023).

Table 1: Experiment results on code generation benchmarks. We report Pass@1 as evaluate metric. Results from the original paper are underlined, and the best results are bold.

| Model | Method | HumanEval | | MBPP | | xCodeEval | | | | |
|---|---|---|---|---|---|---|---|---|---|---|
| | | Pass@1 | Δ↑ | Pass@1 | Δ↑ | *Easy* | *Mid* | *Hard* | *Expert* | All |
| GPT-3.5 | Standard | 68.3 | - | 72.0 | - | 44.4 | 15.2 | 4.6 | 0.0 | 20.2 |
| | CodeT | 81.1 | +12.8 | 76.0 | +4.0 | 50.6 | 16.1 | 8.0 | 0.0 | 23.2 |
| | Reflexion | 69.5 | +1.2 | 72.5 | +0.5 | 44.4 | 17.0 | 5.7 | 0.0 | 20.6 |
| | LDB | 82.9 | +14.6 | 76.0 | +4.0 | - | - | - | - | - |
| | FUNCODER | **85.4** | +17.1 | **78.5** | +6.5 | **62.4** | **29.5** | **11.6** | 0.0 | **31.4** |
| GPT-4 | Standard | 82.9 | - | 73.5 | - | 68.5 | 39.3 | 19.5 | 1.7 | 37.4 |
| | Parsel | 85.0 | +2.1 | - | - | - | - | - | - | - |
| | CodeT | 90.9 | +8.0 | 77.0 | +3.5 | 76.4 | 51.8 | 21.8 | **3.4** | 44.0 |
| | Reflexion | 91.0 | +8.1 | 77.1 | +3.6 | 71.3 | 41.1 | 19.5 | 2.5 | 38.6 |
| | MetaGPT | 85.9 | +3.0 | - | - | - | - | - | - | - |
| | FUNCODER | **94.5** | +11.6 | **79.5** | +6.0 | **83.1** | **58.0** | **26.4** | **3.4** | **48.6** |
| Llama3$_{8b}$ | Standard | 61.6 | - | 60.5 | - | 9.0 | **1.8** | 0.0 | 0.0 | 3.6 |
| | CodeT | 68.9 | +7.3 | 61.5 | +1.0 | 12.4 | 0.0 | 0.0 | 0.0 | 4.4 |
| | FUNCODER | **79.7** | +18.1 | **62.5** | +2.0 | **22.0** | 0.9 | 0.0 | 0.0 | **8.0** |
| StableCode$_{3b}$ | Standard | 61.0 | - | 51.5 | - | 7.3 | 0.9 | 0.0 | 0.0 | 2.8 |
| | CodeT | 75.0 | +14.0 | 57.5 | +6.0 | 11.2 | 1.8 | 0.0 | 0.0 | 4.6 |
| | FUNCODER | **81.0** | +20.0 | **63.5** | +12.0 | **13.5** | **4.5** | **1.1** | 0.0 | **6.2** |
| CodeLlama$_{34b}$ | Standard | 43.9 | - | 53.5 | - | 2.3 | 0.0 | 0.0 | 0.0 | 0.8 |
| | CodeT | 55.5 | +11.6 | 56.5 | +3.0 | 10.1 | 0.0 | 0.0 | 0.0 | 3.6 |
| | FUNCODER | **66.5** | +22.6 | **58.5** | +5.0 | **10.2** | 0.0 | 0.0 | 0.0 | **3.6** |

## 3.1 Code Generation

We choose three benchmarks for code generation evaluation: (a) HumanEval (Chen et al., 2021) includes entry-level coding questions; (b) MBPP (Austin et al., 2021) contains questions of standard library invocation and programming basics; and (c) xCodeEval (Khan et al., 2023) consists of algorithmic challenges sourced from the competitive programming platform CodeForces.

### 3.1.1 Experiment Setup

**Benchmarks** We adopt the full test set (164 problems) for HumanEval, and sample 200 for MBPP and 500 for xCodeEval, respectively. Following EbTech (2024), we split the xCodeEval into 4 subsets based on problem difficulty: Easy ($\leq$ 1200), Mid (1200-1599), Hard (1600-1999) and Expert ($\geq$ 2000). The evaluation metric for code generation is Pass@1 unless specified.

**Baselines** We compare FUNCODER with standard prompting (Brown et al., 2020), two-stage decomposition method Parsel (Zelikman et al., 2023), self-testing method CodeT (Chen et al., 2023a), self-improvement methods Reflexion and LDB (Shinn et al., 2023; Zhong et al., 2024), and multi-agent developing framework MetaGPT (Hong et al., 2024). We implement Standard prompting with a 1-shot demonstration. CodeT samples 11 solutions with standard prompting and evaluates them on model-generated tests. The results for Reflexion are reproduced from the original code.

**Implementation Details** FUNCODER uses a 2-shot prompt in the divide stage and 1-shot for conquering sub-functions. The number of sampled implementations in the functional consensus is set to 11 for code generation tasks. For further implementation details, please refer to Appendix A.1.

### 3.1.2 Results

Table 1 shows the code generation performance on advanced proprietary models, GPT-3.5 (Ouyang et al., 2022) and GPT-4 (OpenAI, 2023). For basic programming questions, HumanEval and MBPP, FUNCODER surpass previous SOTA methods by +3.3% in Pass@1 and reduce the error rate by 18.6%. Furthermore, FUNCODER demonstrates a substantial improvement on competition-level problems, outperforming others by 10.4% in GPT-4 and 35.3% with GPT-3.5. We observe that FUNCODER can

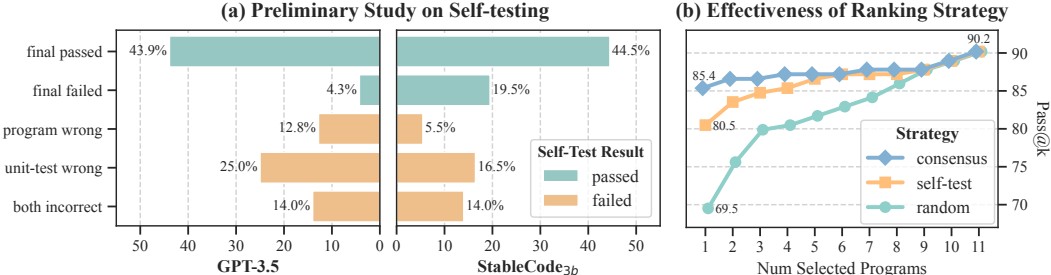

Figure 3: (a) Preliminary study on self-testing, the programs are evaluated using unit-tests generated by LLMs. (b) The effectiveness of different ranking strategies. We compute the Pass@k over top-k programs ranked by functional consensus, self-test, and random on 11 candidates. (higher is better)

enhance LLM's capability of solving more complex programming tasks, with an average accuracy improvement of 82.3% over the baseline on the *Mid* and *Hard* subsets of xCodeEval. *Expert* level programs, however, still remain a colossal challenge for even the most cutting-edge LLMs.

Evaluation is also performed over community LLMs, Llama3 (Meta AI, 2024), StableCode (Pinnaparaju et al., 2024) and CodeLlama (Rozière et al., 2023) with results in Table 1. FUNCODER consistently boosts the performance of smaller models in code generation, demonstrating notable improvements compared to standard prompting on HumanEval, which gained +29.4% on Llama3, +32.8% on StableCode, and even +51.5% on CodeLlama, outperforming that from the previous best method CodeT. We also supplement results on GPT-4o mini, Codestral and StarCoder2 in Table 11. Experiment results demonstrate that our method archives state-of-the-art performance on various models, ranging from basic programming to competition contests.

### 3.1.3 Analysis

**FUNCODER Democratize to Smaller LLMs** Limited by the LLM capabilities, the application of self-improvement or multi-agent methods on smaller models is without ease. By keeping decomposition and composition within the code generation process, our approach exhibits better generalization. As shown in Table 1, with FUNCODER, StableCode$_{3b}$ achieves around $118.6\%$ relative performance to standard GPT-3.5, and also aligns closely with GPT-4 by about $97.7\%$ on HumanEval.

**Preliminary Study on Self-Testing Method** We conduct a preliminary study targeting the self-testing method on HumanEval, results are shown in Figure 3.a with further details in Appendix A.5. We first verify whether model-generated programs can also pass model-generated self-tests: (a) If a program passes self-tests, most from GPT-3.5 would also work on system tests, as much as $^{19.5\%}/_{64\%} \approx 30.5\%$ programs from StableCode are rejected, indicating that smaller models like StableCode may not effectively self-test and detect program errors on its own. (b) In the event of failed self-tests, a large portion of failures are attributed to issues in self-tests instead of the programs, on both GPT-3.5 and StableCode. These phenomena indicate that self-testing methods have limitations in generating correct and reliable unit tests. As a result, we design functional consensus to not require any assertion, but perform *mutual verification* between solutions instead, as opposed to self-testing.

**Effectiveness of Functional Consensus** Functional consensus or self-testing may be viewed as ranking algorithms for selecting functions. To measure ranking effectiveness, we conduct an analysis on HumanEval with GPT-3.5. For each problem, 11 candidates are ranked with 3 strategies: consensus, self-test, and random shuffle (as a baseline). Effectiveness is measured via Pass@k, i.e. if any of the top-k ranked programs pass the system test. Figure 3.b shows that functional consensus achieves $94.7\%$ upper bound (Pass@11) performance by selecting *a single* function (Pass@1), and is close to that of self-test on Pass@4. This clearly demonstrates that functional consensus can effectively evaluate correctness and pick the most promising implementation on the first attempt.

**Ablation and Token Usage** To analyze the impact of dividing, conquering, and functional consensus in FUNCODER, we carry out an ablation study with different settings. Studies that replace consensus with self-testing, or with AlphaCode-like (Li et al., 2022) clustering, are also included. The ablation is constructed on HumanEval with GPT-3.5, as shown in Table 2. Note that to generate every program FUNCODER costs only $O(kN)$ tokens, where $k$ is the number of sampled candidates,

Table 2: Ablation study of FUNCODER on HumanEval with GPT-3.5. The setting in our main experiment is highlighted in bold. Tokens are calculated as the sum of prompts and completions.

| Setting | Divide | Conquer | Ranking | Pass@1 | Avg. Tokens |
|---|---|---|---|---|---|
| Standard | ✗ | ✗ | ✗ | 68.3 | **886.7** |
| One-pass | ✓ | ✗ | ✗ | 72.6 (+4.3) | 1233.7 |
| Two-pass | ✓ | ✓ | ✗ | 78.7 (+10.4) | 3343.2 |
| Two-pass + ST@11 | ✓ | ✓ | Self-Test@11 | 80.5 (+12.2) | 5408.3 |
| Two-pass + CL@11 | ✓ | ✓ | Clustering@11 | 75.0 (+6.7) | 5070.7 |
| FUNCODER@5 | ✓ | ✓ | Consensus@5 | 83.5 (+15.2) | 4040.9 |
| **FUNCODER@11** | ✓ | ✓ | Consensus@11 | **85.4 (+17.1)** | 5402.0 |

and $N$ is the token length of the final program. This is further exemplified and explained in §A.7. We observe that function decomposition and re-composition deliver cumulative performance improvements. Functional consistency is also shown to prevail over self-testing. Putting them all together, FUNCODER received a +17.1 improvement with just $5.09\times$ more tokens over baseline. Compared to previous SOTA LDB ($\approx$ 23K tokens), we are able to gain +2.5 in performance with 76.5% token usage reduction.

## 3.2 Mathematical Reasoning

Code can be viewed as a tool for augmenting the reasoning capabilities of LLMs (Chen et al., 2023b). Alternative to text-based reasoning like Chain-of-Thought (Wei et al., 2022), programs can offer unique advantages in terms of iteration and calculations. To test the generalizability of FUNCODER beyond algorithm challenges, we conduct an experiment on MATH (Hendrycks et al., 2021b), a competition-level mathematical reasoning benchmark.

### 3.2.1 Experiment Setup

**Benchmark**  The experiment is conducted on a subset of the MATH test set, including 500 randomly sampled problems that can be classified into 7 disjoint subjects or 5 difficulty levels. It can be noticed that labels in MATH are formatted in LaTeX, rendering exact-match verdicts impractical. We, therefore, follow previous work (Zhang et al., 2024) and adopt GPT-4 to determine the correspondence between predictions and labels, with further details provided in Appendix A.4.

**Baselines**  We compare FUNCODER with the text-based baselines: Standard Prompting and Chain-of-Thought (Wei et al., 2022), and program-aided baselines: Program-of-Thought (Chen et al., 2023b), Self-Refine (Madaan et al., 2023), Cumulative Reasoning (Zhang et al., 2024). The results of Cumulative reasoning are reported in the original paper. Standard prompting and chain-of-thought reasoning use 7-shot demonstrations constructed from the train set. Program-of-Thought and Self-Refine prompt the model with 1-shot demonstration to generate a `solution()` function that solves the problem. Additionally, self-refine iteratively refines programs based on runtime feedback. All baseline methods are run with self-consistency (Wang et al., 2023) at 5.

**Implementation Details**  FUNCODER adopts a program-aided reasoning setting that writes a `solution()` function and obtains the final prediction by running this program. The number of sampled implementations $|F|$ in functional consensus is set to 5 to match baseline methods.

### 3.2.2 Results

The experimental results on MATH are shown in Table 3. It shows that program-aided reasoning generally outperforms text-based reasoning. With GPT-4 as the backbone, FUNCODER outperforms the strongest baseline Cumulative Reasoning (Zhang et al., 2024) by (6.0 / 8.3%) and surpasses the vanilla program-aided baseline PoT (Chen et al., 2023b) by (10.0 / 14.7%). When using GPT-3.5-turbo as the backbone, FUNCODER exceeds the strongest baseline by (6.2 / 11.1%) and outperforms PoT by as much as (13.0 / 31.7%), which indicates that our approach has a strong advantage over both text-based reasoning and other program-aided reasoning methods.

Table 3: Experimental results on MATH, a competition-level mathematical reasoning benchmark. Best results are in bold. Text-based reasoning methods are denoted with [†], while others use program-aided reasoning. We report both overall results and results in seven subjects: *Prealgebra*, *Algebra*, *Number Theory*, *Counting & Probability*, *Geometry*, *Intermediate Algebra*, and *Precalculus*.

| Model | Method | Prealg. | Alg. | NT | Prob. | Geo. | InterAlg. | Precalc. | Overall |
|---|---|---|---|---|---|---|---|---|---|
| GPT-3.5 | Standard[†] | 62.2 | 37.4 | 20.0 | 29.8 | 31.0 | 24.4 | 21.8 | 34.6 |
| | CoT[†] | 59.8 | 51.1 | 28.9 | 29.8 | 28.6 | 26.7 | 30.9 | 40.0 |
| | PoT | 68.3 | 50.4 | 33.3 | 48.9 | 21.4 | 18.2 | 29.1 | 41.0 |
| | Self-Refine | 74.4 | 49.6 | 48.9 | 57.4 | 28.6 | 35.6 | 36.4 | 48.6 |
| | FUNCODER | **76.8** | **61.2** | **55.6** | **59.6** | **34.1** | **36.0** | **41.8** | **54.0** |
| GPT-4 | Standard[†] | 81.7 | 82.7 | 71.1 | 72.3 | **59.5** | 46.7 | 47.3 | 68.2 |
| | CoT[†] | 84.1 | 87.1 | 62.2 | 68.1 | 45.2 | 48.9 | 54.5 | 68.6 |
| | PoT | 79.3 | 80.6 | 75.6 | 72.3 | 50.0 | 47.8 | 58.2 | 68.2 |
| | Self-Refine | 82.9 | 82.0 | 77.8 | 76.6 | 54.8 | 55.6 | **63.6** | 72.2 |
| | CR | 86.6 | 86.3 | **88.7** | 71.1 | 53.7 | 51.5 | 51.8 | 72.2 |
| | FUNCODER | **89.0** | **92.8** | 82.2 | **83.0** | 59.5 | 63.3 | 56.4 | **78.2** |
| Llama3$_{8b}$ | CoT[†] | 56.1 | **47.5** | 31.1 | 34.0 | **40.5** | 14.4 | **38.2** | 38.6 |
| | PoT | 67.1 | 32.4 | 24.4 | 34.0 | 16.7 | 21.1 | 18.2 | 32.6 |
| | FUNCODER | **67.9** | 45.7 | **51.1** | **53.2** | 19.0 | **37.8** | 30.9 | **45.0** |
| StableCode$_{3b}$ | PoT | 20.7 | 14.4 | 17.8 | 25.5 | **4.8** | 8.9 | 9.1 | 14.4 |
| | FUNCODER | **46.3** | **30.2** | **20.0** | **29.8** | 4.8 | **20.0** | **18.2** | **26.6** |
| CodeLlama$_{34b}$ | PoT | 35.5 | 26.1 | 15.0 | 16.7 | 0.0 | 5.5 | 33.3 | 15.2 |
| | FUNCODER | **44.8** | **46.1** | **37.8** | **34.1** | **13.6** | **24.6** | **37.5** | **24.4** |

On open-source models, FUNCODER with Llama3 outperforms PoT by (12.4 / 38.0%). It has even reached competitive performance against the state-of-the-art method based on GPT-3.5 (45.0 v.s. 48.6). When employing StableCode and CodeLLaMA as the backbone, our approach achieves significant improvements by (12.2 / 84.7%) and (9.2 / 60.5%), respectively. This improvement demonstrates that our approach can significantly boost smaller LLMs, democratizing the complex reasoning capabilities of open-source LLMs through programming.

### 3.2.3 Analysis

**FUNCODER Can Handle Harder Questions**
Figure 4 compares between CoT, PoT, and FUN-CODER across varying difficulty levels. It illustrates that CoT performs comparatively well on the easiest questions, but suffers from a steep decline in performance as difficulty increases. This suggests that text-based reasoning is inadequate for tackling challenging mathematical reasoning problems. The same situation is also observed in PoT. In contrast, our method consistently demonstrates high performance even on challenging problems, particularly excelling on level 5 difficulty with nearly double the performance compared to PoT and CoT. This reflects that our method, with divide-and-conquer applied, can effectively cope with complex problems.

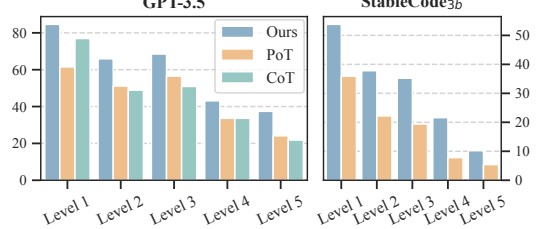

Figure 4: Average accuracy in each level with the chat model (GPT-3.5) and the code model (StableCode$_{3b}$) on the MATH benchmark.

**Decomposed Functions are Domain-Specific** We hypothesize that questions from the same subject require similar knowledge reserves, which should be reflected in the functionality of the sub-functions. To verify this hypothesis, we statisticize the common sub-functions of FUNCODER in each MATH subject, as shown in Table 4. It is apparent that different subjects require different abilities, each with its own set of sub-functions closely associated with the domain knowledge. In addition, these common sub-functions are fundamentally basic and straightforward. As exemplified in Appendix B.2, our method is able to leverage and combine these basic sub-functions to achieve more complex goals, thereby reducing the complexity of reasoning and enhancing performance.

Table 4: Top-3 most commonly used functions in each subject of MATH, listed in descending order.

| Subject | Functions |
|---------|-----------|
| Prealgebra | is_prime / factorial / gcd |
| Algebra | find_roots / is_perfect_square / find_domain |
| Number Theory | get_divisors / mod_inverse / gcd |
| Counting & Probability | factorial / combinations / binomial_coefficient |
| Geometry | distance / simplify_fraction / calculate_triangle_area |
| Intermediate Algebra | find_roots / evaluate_polynomial / lagrange_interpolation |
| Precalculus | cross_product / fraction_from_angle / dot |

## 4 Related Work

**Large Language Model for Code** Code pre-training has received widespread attention, with early models based on small language models (SLM) (Feng et al., 2020; Lu et al., 2021; Wang et al., 2021). In recent years, with the development of large-scale pre-training techniques, code LLM has emerged, showing remarkable performance in downstream code tasks (Chen et al., 2021; Nijkamp et al., 2023; Li et al., 2022; Rozière et al., 2023; Li et al., 2023b; Guo et al., 2024). Tasks between code and natural language (NL) can be generally divided into three major categories: NL2Code tasks such as code generation (Austin et al., 2021; Chen et al., 2021; Hendrycks et al., 2021a; Khan et al., 2023) and code search (Husain et al., 2019); Code2Code tasks including code completion (Lu et al., 2021; Zhang et al., 2023; Liu et al., 2024), code translation (Ahmad et al., 2023; Zhu et al., 2022; Yan et al., 2023), and test generation (Siddiq et al., 2023; Schäfer et al., 2024); Code2NL tasks like code summarization (Jin et al., 2023). This paper focuses on code generation tasks, ranging from basic to competition level.

**Code Refinement and Self-Testing** Code doesn't always run as expected; it could contain syntax errors, dead loops, or bugs. It's essential to debug and refine the code to ensure better quality. CodeT (Chen et al., 2023a) generates unit-tests to score the implementation. AlphaCode (Li et al., 2022) clusters programs based on whether generated program outputs were identical or not. Self-improvement methods (Madaan et al., 2023; Shinn et al., 2023; Chen et al., 2024; Zhong et al., 2024) design closed-loop procedures that repeatedly refine the code based on the feedback. Like real-life software development processes, multi-agent frameworks (Hong et al., 2024; Qian et al., 2023) construct specific LLM roles, *Tester* or *QA* to generate tests. These studies adopt a shared paradigm wherein self-tests are generated through LLMs. However, Olausson et al. (2024) points out the challenge that LLMs have certain shortcomings in self-repairing their code. This paper avoids these shortcomings by proposing *functional consensus* as a reliable method of evaluation.

**Program-Aided Reasoning and Agents** Aside from code generation tasks, the program can be a tool that augments LLM to solve complex reasoning questions or interact with external environments. Program-of-Thought (Chen et al., 2023b) and PAL (Gao et al., 2023) prompt the model to generate a program that solves mathematical or symbolic problems. MathPrompter (Imani et al., 2023) and Chain-of-Code (Li et al., 2023a) fuse the text-based chain-of-thought with code-based program-of-thought prompting to complement each other in mathematical reasoning. Cumulative Reasoning (Zhang et al., 2024) conducts bottom-up reasoning to derive the final answer progressively. Numerous work (Sun et al., 2023; Wang et al., 2024; Yang et al., 2024) also use code as an intermediate component to bridge LLM agents with external environments.

**Decompose for Complex Problems** Several recent works employ decomposition to reduce the complexity of hard problems. Least-to-Most (Zhou et al., 2023) adopts a two-stage approach, which first decomposes complex problems, and then solves each sub-problem individually to tackle complex reasoning tasks. Successive Prompting (Dua et al., 2022) adopts a dynamic decomposition, iteratively breaking down problems and addressing sub-problems. Tree-of-Thought (Yao et al., 2023) breaks down complex problems into state spaces and uses tree search to solve them. Parsel (Zelikman et al., 2023) introduces decomposition to code generation tasks, taking a three-stage to break down requirements into draft and intermediate parsel programs. RepoCoder (Zhang et al., 2023) performs a retrieval in repositories to complete unfinished code one by one. Unlike these methods, FUNCODER recursively decomposes problems into a tree structure, hence gradually reduces its complexity.

# 5    Discussion

**Limitations**  Our approach unleashes the potential power of functions in programming, which is advantageous on well-defined problems such as competitive programming, or program-augmented reasoning tasks. These scenarios do not however represent all use cases, such as open-ended problems or casual software development. Nevertheless, we believe that the idea of divide-and-conquer and sub-modular consensus utilized by FUNCODER can be extended to a wider range of problems, and we consider this as a future exploration.

**Broader Impact**  While code generation is increasingly utilized in software development, Large Language Models (LLMs) are still prone to generating toxic, vulnerable, or malicious code. Such programs pose risks and should be used or executed with extra caution.

# 6    Conclusion

In this paper, we presented FUNCODER, a novel code generation framework that integrates the divide-and-conquer strategy with functional consensus to address complex requirements. FUNCODER had demonstrated superior performance compared to state-of-the-art methods on various benchmarks and models. Our findings highlighted the effectiveness of dynamic decomposition and functional consensus in writing complex code, which suggests that FUNCODER may have the potential to empower further improvements in code generation and other fields.

# Acknowledgments

We would like to acknowledge the reviewers and chairs for their inspiring and constructive feedback. The research in this article is supported by the National Key Research and Development Project (2021YFF0901602), the National Science Foundation of China (U22B2059, 62276083). Ming Liu and Qianglong Chen are the corresponding authors.

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

# A   Appendix

In the supplementary materials, we provide the details of implementation (A.1), baseline information and settings (A.2), benchmarks (A.3), metrics (A.4), settings in the analysis (A.5), and additional experiments (A.9). We also demonstrate the example solutions of our method and baseline in Appendix B, and include all the prompts in Appendix C.

Table 5: Symbols and Glossary.

| | Alias | Description |
|---|---|---|
| *(i) Symbols* | | |
| $f(x)$ | Function | In the programming language, a function consists of header, documentation, and its body $\{h_f, d_f, b_f\}$. A function can also be viewed as a mapping $f : D(f) \rightarrow Y$. |
| $h_f$ | Function Header | Declares the function name, arguments, and return type, and is used as a signature to identify the function in a program. |
| $d_f$ | Function Docstring (or Documentation) | Provides additional usage details for this function, but is optional. We encourage the model to generate docstrings to describe sub-goals precisely. |
| $b_f$ | Function Body (or Implementation) | The function body contains a subroutine that describes its control flow and behavior. Functions may be invoked from within. |
| $f'(x)$ | Partially Implemented | A provisional function structure generated by the LLM where sub-procedures are not yet implemented. |
| $f^*(x)$ | Solved Function | A final implementation that is no longer changed and represents FUNCODER's final comprehension and solution on the original problem. |
| $F = \{f_{(i)}\}$ | Sampled Implementation | Functions that re-implement $f'(x)$ based on solved sub-functions, generated by models using the same input prompt. |
| CHILD$(f(x))$ | Dependency | Functions that are used in $f(x)$. (exclude $f(x)$ itself) |
| $T$ | Dependency Tree | Defined by TREE$(f, \text{CHILD}(f))$, where $f$ is the root node of the current sub-task. Circular references are ignored. |
| $\mathcal{F}$ | Function Composition | To implement a certain function $f$ respecting sub-procedures as potentially reusable components. |
| *(ii) Glossary* | | |
| System Test | Hidden Test | System testing is a phase where a set of previously invisible test cases are run against the submitted program to validate if the code is correct and produces the expected output for different categories of inputs. |
| Unit Test | Assertion | A unit test is an assertion consisting of given input and expected output, whereas in Python, it takes the form of `assert func(x) == y`. |
| Self-testing | - | Self-testing is an evaluation process that prompts the model to generate unit tests (assertions) to assess the correctness of the generated program. |
| AlphaCode-like Clustering | - | AlphaCode proposed a clustering process that elects candidate program from a number of samples, recognizing programs that produce exactly identical outputs as equivalent, and picks one program from the largest cluster. |

## A.1   Implementation Details

**Models**   We access the OpenAI models GPT-3.5 (gpt-3.5-turbo-0613), GPT-4 (gpt-4-1106-preview) and GPT-4o mini (gpt-4o-mini-2024-07-18) through Azure OpenAI. Weights of community models Llama3 (Meta-Llama-3-8B-Instruct), Codestral (Codestral-22B-v0.1), StableCode (stable-code-instruct-3b), CodeLlama (CodeLlama-34b-Instruct-hf) and StarCoder2 (starcoder2-15b-instruct-v0.1)

are downloaded from HuggingFace (Wolf et al., 2019) and served over an OpenAI-like API on a single A100-80G GPU under BF16 precision with vLLM (Kwon et al., 2023).

**Divide**   We instruct the model to write the current function and introduce new functions with clearly defined sub-goals. The prompt C.2 for the *divide* process includes two examples: one example needs to involve new functions that are left unimplemented; and another where the sub-goal is simple enough that no further decomposition is necessary. The model generates a Python code block with a temperature of 0.2, and the code block will be extracted to represent a tree of functions with new functions as the children of the current. We require that any new sub-function do not refer to existing functions, to avoid circular references. This generation process will be attempted at most 3 times until any valid code with a proper signature is found in the output. FUNCODER then traverses the function tree via depth-first search and restricts the max depth of the tree to 6.

**Conquer**   We apply the composition of sub-functions to rewrite the parent function after all sub-functions have been fully decomposed. Code for sub-functions is made visible to the LLM, which is requested to rewrite the current function with a 1-shot demonstration (C.3). With functional consensus applied, the model samples multiple implementations with a temperature of 0.8, and the one that reaches consensus will be kept for further bottom-up processing.

**Functional Consensus**   The functional consensus is applied in the conquer stage. Formally, Consensus@k samples k-1 implementations in the conquer stage, and reuses the one produced in the divide stage, resulting in a set $F$ of k candidate programs. Then we prompt the model with 1-shot (C.4) to generate potential inputs $X$ for the given function and use them to feed and execute the program. As described in Eq 2, when two functions output the same value in a given input, they will both add 1 point to the overall similarity. A thrown exception or timeout during execution assigns -100 points to the candidate as it indicates potentially problematic code. Similar to self-testing methods, we also leverage the example input/output at the root node to filter out candidates that have wrong functionality. Finally, the one candidate with maximum scores over all inputs is selected, as it reaches consensus with other implementations.

**Hierarchical Code Interpreter**   Divide-and-conquer represents the problem hierarchy through structured code. To gain insights of this information, we design an interpreter that syntactically parses the generated output and organizes them into a graph of functions. We are thus able to decompose complex tasks by representing sub-goals through the connections of multiple functions. LLMs may produce vulnerable code even if prompted by trusted inputs, making direct execution or `eval()` on generated code especially hazardous. Our framework addresses this with the use of a sandboxed environment to contain untrusted code execution, preventing the LLM from hanging up or even breaking the system.

### A.2   Baseline Details

**Standard Prompting**   conducts one-time generation and directly output the entire code or final results. In code generation tasks, we use a 1-shot prompting setting with 0.3 temperature. For MATH, we sample 1 question-answer pair per subject in the train set, resulting in a 7-shot prompt, and run self-consistency (Wang et al., 2023) with consistency@5 and temperature 0.7.

**CodeT** (Chen et al., 2023a) samples multiple code solutions $X$ and unit-tests $Y$. A unit test is an assertion consisting of given input and expected output, whereas in Python it takes the form of `"assert func(x) == y"`, CodeT then checks the programs over self-tests and divides the functions into sets; the score of such a set is defined as the number of functions within multiplied by the number of succeeded tests. Finally, CodeT selects the function with the most agreement (in the biggest set). Similar to the setting of FUNCODER, we sample 11 candidate solutions with 0.8 temperature.

**AlphaCode-like Clustering**   is introduced with the model AlphaCode (Li et al., 2022), and samples multiple code solutions $X$. A fine-tuned model is used to generate test inputs upon which programs are evaluated. Programs are then clustered by whether the outputs are (exactly) identical, in which the largest group was selected. The final result comes from this largest group, where any result within would suffice. In our ablation study, we use similar settings to FUNCODER, sampling 11 candidate solutions with 0.8 temperature and generate sample inputs likewise.

**Parsel** (Zelikman et al., 2023) consists of three generation stages: high-level sketch, Parsel program, and final program. The Parsel program is an intermediate representation of code that describes and organizes program structure. We report the result of HumanEval with GPT-4 from the original paper.

**Reflexion** (Shinn et al., 2023) is a closed-loop agent system that generates unit tests and iteratively refines the program based on the self-test feedback. The results for GPT-4 on HumanEval and MBPP are reported in the original paper. Based on officially released code[3], we test results with GPT-3.5 and community models under the reflexion strategy with `max_iters=2` and Pass@1. For the xCodeEval benchmark, as it is judged through standard input/output, we wrap the standard input into function arguments and obtain the return value as the output in the form of `"def main(input_str:  str) -> str"`, and the sample input/output are also transformed to visible tests for reflexion process.

**MetaGPT** (Hong et al., 2024) employs a multi-agent strategy that assigns roles and encodes human-like software development procedures. The scripts for reproducing the results were not made public as of this paper was completed. Therefore, we include the original result for GPT-4 on the HumanEval dataset under the with feedback setting.

**LDB** (Zhong et al., 2024) segments programs into basic blocks and tracks the values of intermediate variables after each block throughout runtime execution, allowing large language models to verify the correctness of smaller code units. We adopt the results as-is reported in the paper.

**Chain-of-Thought Prompting** (Wei et al., 2022) generates step-by-step reasoning leading to the final output answer. The solution is formatted in LaTeX, and use \boxed to mark the final answer. We sample 1 shot per subject in the MATH train set, resulting in a 7-shot demonstration, and running with consistency@5 and a temperature of 0.7.

**Program-of-Thought** (Chen et al., 2023b) utilizes the coding ability in LLMs to generate programs rather than text-based solutions for reasoning tasks. In MATH, we hint the model with 1-shot prompting to generate a `solution()` function that returns the final answer to the problem. The program is then executed in a Python environment and obtains the return value. If an exception is thrown during execution, the model will try to regenerate a new program until it succeeds or reaches 3 attempts. Similar to CoT, Program-of-Thought samples 5 programs at a temperature of 0.7 and votes the final result.

**Self-Refine** (Madaan et al., 2023) iteratively prompts the model to give feedback and refine the generated code based on it. Self-refine does not incorporate self-tests, and the refinement is conducted solely on model feedback. In our preliminary study on HumanEval, this feedback is weak and cannot improve performance. However, in MATH, the solution program can be executed without the need for generated assertions. Thus, we extend the self-refine to capture the runtime error trace as feedback and refine the code until it can run or exceed 3 retries.

**Cumulative Reasoning** (Zhang et al., 2024) starts from decomposing the input problem into propositions and conducts bottom-up reasoning until the final answer can be concluded. The results for Cumulative Reasoning are reported in the original paper under with code setting.

### A.3    Benchmark Details

Table 6: Overview and details of HumanEval, MBPP, xCodeEval, and MATH dataset.

| | **HumanEval** | **MBPP** | **xCodeEval** | **MATH** |
|---|---|---|---|---|
| Task Attribute | Code Generation - | Code Generation - | Programming Contest tags, difficulty | Mathematical Reasoning subject, level |
| Metric | Pass@1 | Pass@1 | Pass@1 | EM-gpt |
| # Sample (original) | 164 | 427 | 7,635 | 5,000 |
| # Sample (ours) | 164 | 200 | 500 | 500 |
| Entry func | variant | variant | main() | solution() |
| Input | arguments | arguments | standard input | n/a |
| Output | return | return | standard output | return |
| # Examples Tests | ~2.8 | 0 | ~2.1 | n/a |
| # System Tests | ~8.1 | ~3.1 | 51.1 | n/a |

---

[3]GitHub: noahshinn/reflexion

**HumanEval** (Chen et al., 2021) is a hand-crafted programming dataset designed to evaluate a model's code generation capability. It consists of 164 instances involving programming skills in language comprehension, reasoning, algorithms, and simple mathematics. The problem contains 2.8 sample inputs and outputs on average in the function document, which can be leveraged to provide additional guidance for the LLM to select or self-improve the programs. We conduct experiments on all 164 instances using accuracy (Pass@1) as the evaluation metric. The details of the Pass@1 metric are described in Appendix A.4.

**MBPP** (Austin et al., 2021) consists of fundamental Python programming problems, with a total of 974 examples covering Python programming basics, standard library usage, and related assessment. Following Shinn et al. (2023), we adopt the mbpp-typed split from MultiPL-E (Cassano et al., 2023) and sample 200 instances, using Pass@1 as the metric. The original prompt[4] from MBPP includes all hidden tests in the input problem, which may cause label leakage when using these tests to refine or select programs. To ensure a fair comparison, MultiPL-E removes the test information in the prompt.

**xCodeEval** (Khan et al., 2023) is a competition-level multilingual and multitask benchmark consisting of 17 programming languages. xCodeEval collects 25 million openly available samples from codeforces.com, a platform for competitive programming. The data we use include problem descriptions in `problem_descriptions.jsonl` and system tests from `unittest_db.json` which consists of 7,635 competition problems and averaged 51.1 tests per problem. Note that the tests in xCodeEval are crawled, some of them are incomplete due to the website context limit (they end with an ellipsis and the further content is missing); we filter out problems having invalid test cases. Based on the CodeForces Rating (EbTech, 2024), we categorize the problems by their difficulty: Easy ($\leq$ 1200), Mid (1200-1599), Hard (1600-1999), and Expert ($\geq$ 2000). We sample 500 problems from the full split with the basic filter rule mentioned above, resulting in Table 7. The CodeForces problem has a different input/output style compared to HumanEval and MBPP; it scans input from Standard Input and prints the answer to Standard Output. Therefore, we judge the program on system tests using a CodeForces-style judger and use Pass@1 (the program must pass all system tests) as the evaluation metric.

**MATH** (Hendrycks et al., 2021b) is a challenging competition-level mathematical reasoning dataset, with problems and solutions formatted in LaTeX. It covers seven categories: Prealgebra, Algebra, Number Theory, Counting & Probability, Geometry, Intermediate Algebra, and Precalculus. The original test set of MATH consists of 5000 samples, and we randomly sampled 500 problems as shown in Table 7. In addition to text-based reasoning, writing programs is another promising way to solve mathematical problems. These methods involve writing a `main()` or `solution()` function, and executing the program to obtain the final answer. Through experiments on MATH, we aim to demonstrate that FUNCODER can enhance LLM's ability to address complex mathematical problems through programming.

Table 7: Number of test samples in **(a)** xCodeEval difficulty, **(b)** MATH level, **(c)** MATH subject.

| Difficulty | Ours | Original |
|---|---|---|
| Easy | 178 | 1428 |
| Mid | 112 | 1319 |
| Hard | 87 | 1453 |
| Expert | 118 | 3289 |
| n/a | 5 | 146 |
| Total | 500 | 7635 |

| Level | Ours | Original |
|---|---|---|
| Level 1 | 39 | 437 |
| Level 2 | 90 | 894 |
| Level 3 | 108 | 1131 |
| Level 4 | 116 | 1214 |
| Level 5 | 147 | 1324 |
| Total | 500 | 5000 |

| Subject | Ours | Original |
|---|---|---|
| Prealgebra | 82 | 871 |
| Algebra | 139 | 1187 |
| Number Theory | 45 | 540 |
| Counting & Probability | 47 | 474 |
| Geometry | 42 | 479 |
| Intermediate Algebra | 90 | 903 |
| Precalculus | 55 | 546 |
| Total | 500 | 5000 |

## A.4 Metrics

**Pass@k** When a program is passed (or accepted), it means that the program must pass all system tests without errors and within the time limit. In our experiments, we set the time limit to 2.5 seconds. Pass@k judges k independent programs, and if any of them can pass, the result will be 1.

---

[4]Original MBPP prompt: https://github.com/google-research/google-research/tree/master/mbpp

In most of our experiments, we use Pass@1 as the metric, as it reflects the accuracy of the method framework achieved without feedback from humans. Pass@k, on the other hand, is equivalent to filtering programs through hidden, human-annotated test labels.

**EM-GPT**   The ground truth label in MATH is written in LaTeX, and the accuracy between labels and model predictions cannot be directly calculated through exact-match (EM). MATH provides a judge program[5] that preprocesses LaTeX syntax and check whether two disambiguated strings are equal. However, this is insufficient for evaluating LaTeX-formatted labels with variant program outputs. We follow the evaluation criteria from previous work (Zhang et al., 2024), using GPT-4 to assess the consistency between predictions and ground truths, with prompt shown in C.6.

## A.5   Details of Analysis

**Details of Preliminary Analysis on Self-testing**   (Figure 3.a) The preliminary study is conducted on the HumanEval dataset, which includes system tests $S$ to evaluate the accuracy of the program, as well as one human-annotated canonical solution $c$. For each question, we: (1) Obtain one solution program $p$ from Standard Prompting. (2) Prompt the model to generate 7 self-tests $T$ based on the question and entry function. The self-test is in the form of the unit test `assert f(x) == y`. We then judge the generated program $p$ and canonical solution $c$ over the self-tests $T$ and system tests $S$. Formally, a pair $(x, Y)$ is used to identify whether program $x$ passes test $Y$. Where $(p, S)$ indicates that the program can pass the system tests, demonstrating its correctness. And $\neg(c, T)$ means the canonical solution can not pass self-tests, suggesting that the tests generated by model could be wrong. The self-test results on generated programs are first evaluated and divided into two classes: self-test passed or failed. If the self-test passes, the self-improvement methods will stop iteration and pick this program as a final result. The next step is to determine whether the program can pass system tests. If the self-test fails, it indicates that there could be an error in the program or test itself. In this case, the correctness of the program is checked using final tests $(p, S)$ and the correctness of the unit test by canonical program $(c, T)$. The results on GPT-3.5 and StableCode are shown in Figure 3 and detailed explanations about these conditions can be found in Table 8.

Table 8: Explanation on how we classify cases in self-testing preliminary study.

| Class | Subclass | Condition | Explanation |
|---|---|---|---|
| self-test passed | final passed | $(p, T) \wedge (p, S)$ | The self-test result is consistent with the final judge. However, self-testing methods **cannot improve** performance in this case, as the program from the baseline (Standard Prompt) is already correct. |
| | final failed | $(p, T) \wedge \neg(p, S)$ | Self-test is **too weak** to detect errors in the program, there could be edge cases that not been considered. |
| self-test failed | program wrong | $\neg(p, T) \wedge \neg(p, S) \wedge (c, T)$ | This is a **good example** that self-testing detects errors in the program. Feedback from the test will be used to select or refine the solution. |
| | unit-test wrong | $\neg(p, T) \wedge (p, S) \wedge \neg(c, T)$ | Bad case, the self-test produced an **error result** and filtered out a correct solution. Continuously revising the code for this test will lead to a performance downgrade. |
| | both wrong | $\neg(p, T) \wedge \neg(p, S) \wedge \neg(c, T)$ | The model is unable to generate a correct solution or test cases. Refining the program over **faulty test samples** will not lead to the correct answer. |
| | - | $\neg(p, T) \wedge (p, S) \wedge (c, T)$ | In the event of self-test failure, there must have been at least one error in either program or tests, so this condition **should never occur**. |

**Details of Ranking Strategy Comparison**   (Figure 3.b) We obtain 11 candidate programs from FUNCODER on HumanEval with GPT-3.5 and rank them through three strategies. This ensures that the same candidate set is used for a fair comparison. An effective ranking strategy should prioritize

---

[5]math_equivalence: https://github.com/hendrycks/math/blob/main/modeling/math_equivalence.py

placing correct programs at the forefront and filter out those with errors. Thus, we measure the effectiveness by computing Pass@k results on the top-k-ranked programs selected by each strategy. The Pass@11 result serves as an upper bound as it uses all programs to compute the pass rate.

**How We Count Frequently Used Functions in MATH** (Table 4) In the mathematical reasoning experiments, we used a subset of 500 items from the MATH test set, with an average of 71.4 questions per subject. However, it is not very confident to represent common functions from only 71.4 programs. Therefore, we sample 3000 problems from the MATH test set for this experiment and run the *divide-only* setting of FUNCODER on them. Then, the occurrence of sub-functions is counted based on their names after extracting the function nodes of code trees for each category.

### A.6 Detailed Explanation of Algorithm

We hereby provide a detailed explanation of FUNCODER algorithm works, with respect to Algorithm 1 from Figure 2 (a copy is included below for simple reading). As mentioned, FUNCODER is a recursive process following a DFS pattern. We use square brackets (e.g. [L1]) below to denote line numbers in the pseudocode.

**Algorithm 1** FUNCODER procedure

**Require:** Entry func, $f_{\text{root}} = \{h_{\text{root}}, d_{\text{root}}, \phi\}$
**Require:** Large language model, LLM
1: **function** FUNCODER($f_{\text{cur}}$)
2:    — *Divide* —
3:    $f'_{\text{cur}}, \{f_i\} \leftarrow$ EXTRACT(LLM($f_{\text{cur}}$))
4:    **for** $f_i \in \{f_i\}$ **do**
5:       **if** $b_i$ **is** NOTIMPLEMENTED **then**
6:          $f_i^* \leftarrow$ FUNCODER($f_i$)        ▷ recursion
7:       **end if**
8:       ADDCHILD($f_{\text{cur}}, f_i^*$)
9:    **end for**
10:    — *Conquer* —
11:    $F_{\text{cur}} \leftarrow$ SAMPLE(LLM($f'_{\text{cur}}$, CHILD($f_{\text{cur}}$)))
12:    $f_{\text{cur}}^* \leftarrow$ FUNCONSENSUS($F_{\text{cur}}$)
13:    **return** $f_{\text{cur}}^*$
14: **end function**
15: **return** FUNCODER($f_{\text{root}}$)        ▷ starts from root

Figure 5: Left: Algorithm for FUNCODER. Right: Decomposition example of A[B[DE]C].

FUNCODER [L1], when solving each function $f$, first performs the *Divide* stage [L3-L9], where the LLM initially writes the function and identifies some potential sub-problems, represented as sub-function stubs (e.g., def f(xs: list[int]) -> int) [L3]. In this process, we identify the sub-problems of the current problem, thereby understanding the dependency between functions.

For each decomposed sub-problem $g_1, g_2, \ldots$, we recursively use FUNCODER to obtain the final implementation $G_i$ for that sub-problem [L5-L8]. This $G_i$ shall replace the previously incomplete subfunction stub signature in the final program.

Now that all sub-problems of $f$ are implemented, we move on to the Conquer stage [L11-13] to complete the larger problem. By combining the signature $f$ and the final implementations $G_1, G_2, \ldots$ of sub-problems, we generate the complete implementation $F$ [L11] and return it [L13].

We hierarchically describe how this algorithm works in detail by combining it with the example given in the right half of Figure 5.

```
[a.1] FunCoder(a)
|  [a.3] LLM(a) -> A, {b, c}      # divide
|--[b.1] FunCoder(b)
|  |  [b.3] LLM(b) -> B, {d, e}   # divide
```

```
|  |--[d.1] FunCoder(d)
|  |  |  [d.3] LLM(d) -> D, {}      # divide
|  |  +--[d.13] return D            # nothing to conquer
|  |--[e.1] FunCoder(e)
|  |  |  [e.3] LLM(e) -> E, {}      # divide
|  |  +--[e.13] return E            # nothing to conquer
|  |  [b.11] LLM(B, {D, E}) -> B*   # conquer
|  +--[b.13] return B*
|--[c.1] FunCoder(c)
|  |  [c.3] LLM(c) -> C, {}         # divide
|  +--[c.13] return C               # nothing to conquer
|  [a.11] LLM(A, {B, C}) -> A*      # conquer
+--[a.13] return A*                 # final result
```

## A.7 Token-cost Complexity

**Example**   We use the example from Figure 5, where the final program consists of 5 functions `A[B[D,E],C]`, and `A` serves as the entry to the program. Here we respect the aforementioned notations, and further use the lower-case letter $a$ to represent the number of stub tokens, upper-case $A$ to represent the number of result tokens, for the function `A`, and other functions likewise. Let $N = A + B + C + D + E$ be the token-cost complexity of the final result.

In *Standard*, the code is only generated once to complete the given stub. We use parentheses to represent the order of LLM calls in a full process.

```
(1)            a -> A B C D E
   input tokens  = a
   output tokens = A + B + C + D + E
   overall       = O(N)
```

In each step of the FUNCODER/*Divide* stage, the to-be-implemented function will serve as the context. The function (stub) will be implemented and sub-function stubs are to be declared.

```
(1)            a -> A b c
(2)            b -> B d e
(3)            d -> D
(5)            e -> E
(8)            c -> C
   input tokens  = a + b + c + d + e
   output tokens = A + b + B + c + C + d + D + e + E < 2N
   overall       = O(N)
```

In every FUNCODER/*Conquer*, the context shall include the current function's definition and finalized implementations of sub-functions. The output is the re-implemented current function.

```
(4)            d -> kD
(6)            e -> kE
(7)        b D E -> kB
(9)            c -> kC
(10)       a B C -> kA
   input tokens  = a + b + B + c + C + d + D + e + E < 2N
   output tokens = kA + kB + kC + kD + kE = kN
   overall       = O(kN)
```

These stages in all bring FUNCODER's total token consumption to strictly $O(kN)$ for every problem.

**Token Complexity of FUNCODER is $O(kN)$**   Define $N$ as the token length of the final program, which is correlated to the inherent complexity of the problem, and define $k$ as the number of sampled candidates. We first explain in detail that the worst-case token cost of FUNCODER is $O(kN)$:

- The naive *Standard* method should naturally generate $O(N)$ tokens. Sampling-based baselines like *CodeT* cost $O(kN)$ tokens.

- FunCoder goes through the *Divide* stage and the *Conquer* stage for each of the functions.

- Based on the current function, *Divide* generates an implementation of itself and stubs for sub-functions. Within this stage, each function would appear at most once in input and twice in output. All *Divide* stages consume no more than $3N$ tokens.

- *Conquer* regenerates the parent function based on its stub and all finalized sub-functions. Herein each function will appear at most twice in input, and sampled $k$ times in output. If $k = 1$, consensus is implicitly disabled. All *Conquer* stages shall consume at most $(k + 2)N$ tokens.

So FunCoder requires no more than $(k + 5)N$ tokens in input-output, making its token consumption $O(kN)$ even at worst-case, aligning with other sampling-based methods such as CodeT and AlphaCode-like clustering. Furthermore, when sampling is disabled ($k = 1$), our method has a token consumption of $O(N)$, which also aligns with the vanilla *Standard* method.

### A.8 Discussion About Functional Consensus

This section focuses on why *functional consensus* might enhance the correctness of programs and how it differs from other consistency-based methods. Self-consistency (Wang et al., 2023) is widely employed in the realm of LLM reasoning. It samples multiple sets of answers and uses voting to select the most consistent result, where the answers typically consist of named entities, choice options, or numbers. However, this approach faces challenges when voting on sampled programs, as programs describe executable logic instead of data, making it unobvious to determine whether two programs are equivalent just from the looks.

When it comes to picking programs, *functional consensus* in FUNCODER looked beyond the literal symbols and used a different approach. It uses inputs and execution results to compare behavioral differences among programs. There have been similar methods, such as the strict clustering approach in AlphaCode (Li et al., 2022), which samples a set of program inputs and then clustering programs with identical outputs into the same group. The final program is then selected from the largest cluster.

However, the idea of grouping programs by the 'identicalness' of outputs is not without fallacies, since programs rarely specialize in solving one single irreducible problem – they deal with a variety of inputs, conditions and mysterious cases. The result of this, where different solutions could have many common behaviors and some distinct behaviors, is referred to as the term 'special-case similarity' in the FrAngel paper (Shi et al., 2019). We consider a correct program solution which has multiple 'special-case similar' programs that are partially correct in different ways, for example:

- One program behaving correctly on the general case (almost all) but missed a few edge cases

- Another program got one edge case correct but didn't manage to deal with the general case

- Yet another program got all edge cases correct but crashed on the general case

- A buggy program that behaves correctly on all available test cases but none of these tests trigger the bug (literally test coverage problem)

- And many programs that turn out to be frenzy mixtures of all the above

If we had a pool of programs that contained the fully-correct program and an assortment of other programs that respected certain cases of the problem as aforementioned, it'd be obvious that the fully-correct would be decently 'special-case similar' to the rest of the programs, for their similar behavior on inputs. These execution outputs are programmatically obtained and automatically compared against each other without any human intervention or LLM calling required, the process of which sits at the core of our *functional consensus* algorithm.

Therefore, with *functional consensus*, where the solutions with common behaviors are promoted, we could intuitively expect the result to be a higher likelihood of a fully-correct program. Provided below is a hypothetical example demonstrating why *functional consensus* prevails:

**Example** Consider the problem of finding all square roots in the complex domain of a non-zero real number (stored in `float32`). To get the answer right for all inputs, the function must consider 2

cases: A) non-zero numbers have 2 square roots; B) square roots of negative numbers are imaginary. 10 candidate functions are sampled as below:

- 5 results $(a_1, a_2, a_3, a_4, a_5)$ only considered case A and got just positive inputs right. For negative numbers, they literally gave up and crashed.
- 3 results $(b_6, b_7, b_8)$ remembered to consider case B, gave 2 imaginary results for negative numbers, but forgot to do the same for positive numbers, returning only 1 square root therein.
- Only 2 results $(c_9, c_{10})$ considered all cases and returned correct results for all inputs.

If we pick the program through 'clustering', the final result would be one of the 5 results $(a_1, a_2, \ldots, a_5)$ that only considered case A, which is evidently not the correct solution. But with *functional consensus*, the final result is vastly different, since we consider the similarity between the functions based on their behavior on different inputs. Without loss of generality, suppose that there are 2 test inputs $4.0, -9.0$, one for each of the 2 cases. We calculate the similarity as follows:

- Programs $a_i$ got only $\sqrt{4.0} = [2.0, -2.0]$ right so each program here are similar with programs $(a_1, a_2, a_3, a_4, a_5, c_9, c_{10})$, scoring 7 points.
- Since $b_j$ only went well with $\sqrt{-9.0} = [3.0i, -3.0i]$, programs here only score 5 points for case B with the ones $(b_6, b_7, b_8, c_9, c_{10})$.
- Each program in $c_k$ gets 7 points for $\sqrt{4.0}$ with $(a_1, a_2, a_3, a_4, a_5, c_9, c_{10})$, and gets 5 points for $\sqrt{-9.0}$ with $(b_6, b_7, b_8, c_9, c_{10})$. Totals to 12 points.

The final result apparently leaned towards $c_k$ as the correct solution, even if their outputs as a whole weren't even half as much as $a_i$ is. Through this example, we illustrate that *functional consensus* has the potential to identify the correct samples even at their minority, outperforming other methods such as self-consistency or clustering.

### A.9 Supplementary Results

**Token Usage** We provide token usage results in Table 9 for FUNCODER and baseline methods on the HumanEval dataset with the GPT-3.5 model, whereas usage results on other datasets are provided in Table 10. We report the average token usage per problem. The token usage is computed through the sum of prompt tokens and completion tokens returned by OpenAI API chat completion call[6]. For LDB, we report their token usage in the original paper (Zhong et al., 2024).

Table 9: Token usage for different settings of FUNCODER and baseline methods on HumanEval, all evaluated on GPT-3.5-turbo. The LDB results are reported in the original paper. The main setting for LDB and FUNCODER is bolded.

| Method | Setting | Pass@1 | Tokens | | | |
| --- | --- | --- | --- | --- | --- | --- |
| | | | Min. | Max. | Avg. | Med. |
| Standard | One-time | 68.3 | 648 | **1477** | **886.7** | 861 |
| CodeT | One-time + Self-Test@11 | 81.1 (+12.8) | 2298 | 9645 | 4479.1 | 4166 |
| Reflexion | maxiter=2 | 69.5 (+1.2) | **416** | 4906 | 1416.1 | **754** |
| LDB *(reported)* | line-level | 80.5 (+12.2) | - | - | 24K | - |
| | **block-level** | 82.9 (+14.6) | - | - | 23K | - |
| | function-level | 79.9 (+11.6) | - | - | 27K | - |
| FUNCODER | One-pass | 72.6 (+4.3) | 826 | 3489 | 1233.7 | 1132 |
| | Two-pass | 78.7 (+10.4) | 2197 | 8406 | 3343.2 | 3078 |
| | Two-pass + Consensus@5 | 83.5 (+15.2) | 2455 | 9432 | 4040.9 | 3800 |
| | **Two-pass + Consensus@11** | **85.4 (+17.1)** | 3015 | 13850 | 5402.0 | 5166 |
| FUNCODER *(ablation)* | Two-pass + Self-Test@11 | 80.5 (+12.2) | 2967 | 13758 | 5408.3 | 5184 |
| | Two-pass + Clustering@11 | 75.0 (+6.7) | 3044 | 9958 | 5070.7 | 4888 |

---

[6]https://platform.openai.com/docs/guides/text-generation/managing-tokens

Table 10: Token usage of FUNCODER and baseline methods on other datasets, i.e. MBPP, xCodeEval and MATH. Results are evaluated on GPT-3.5-turbo.

| Dataset | Method | Pass@1 | Tokens | | | |
|---|---|---|---|---|---|---|
| | | | Min. | Max. | Avg. | Med. |
| MBPP | Standard | 72.0 | 577 | **2102** | **744.9** | 717.0 |
| | CodeT | 76.0 (+4.0) | 2232 | 8172 | 2945.3 | 2866.0 |
| | Reflexion | 72.5 (+0.5) | **391** | 3379 | 1205 | **569.5** |
| | FUNCODER | **78.5 (+6.5)** | 3462 | 13229 | 5049.9 | 4644.0 |
| xCodeEval | Standard | 20.2 | **1051** | **3343** | **1599.5** | **1530.0** |
| | CodeT | 23.2 (+3.0) | 2264 | 9245 | 3937.4 | 3733.0 |
| | Reflexion | 20.6 (+0.4) | 2977 | 1003222 | 401767.3 | 328591.5 |
| | FUNCODER | **31.4 (+11.2)** | 4883 | 53225 | 10559.7 | 8927.0 |
| MATH | PoT | 41.0 | **551** | **5867** | **953.0** | **835.0** |
| | FUNCODER | **54.0 (+13.0)** | 2622 | 30139 | 7075.5 | 5666.5 |

**Full Results for Code Generation**   We provide results for all conducted experiments on code generation benchmarks in Table 11. Our method consistently improves the baseline on community models by averaging $11\%$ on MBPP and $150\%$ on xCodeEval. It is worth noting that small models have a tendency to have low pass rates on competition problems, leading to a relatively higher randomness, therefore we run 3 experiments and report the median result.

**Full Results for MATH**   The MATH dataset divides the problems into five levels of difficulty. The difficulty distribution of our test set can be found in Table 7. We report the average accuracy of FUNCODER and other methods for each math subject in Table 12 and results for each level in Table 13. The results of Cumulative Reasoning are obtained from the original paper (Zhang et al., 2024). Experiment results demonstrate that our method consistently enhances the model's reasoning ability across all levels of MATH.

Table 11: Results for Code Generation. We report Pass@1 as evaluate metric. Results from the original paper are underlined, and the best results are bold.

| Model | Method | HumanEval | | MBPP | | xCodeEval | | | | |
|---|---|---|---|---|---|---|---|---|---|---|
| | | Pass@1 | Δ↑ | Pass@1 | Δ↑ | Easy | Mid | Hard | Expert | All |
| GPT-3.5 | Standard | 68.3 | - | 72.0 | - | 44.4 | 15.2 | 4.6 | 0.0 | 20.2 |
| | CodeT | 81.1 | +12.8 | 76.0 | +4.0 | 50.6 | 16.1 | 8.0 | 0.0 | 23.2 |
| | Reflexion | 69.5 | +1.2 | 72.5 | +0.5 | 44.4 | 17.0 | 5.7 | 0.0 | 20.6 |
| | LDB | 82.9 | +14.6 | 76.0 | +4.0 | - | - | - | - | - |
| | FUNCODER | 85.4 | +17.1 | 78.5 | +6.5 | 62.4 | 29.5 | 11.6 | 0.0 | 31.4 |
| GPT-4 | Standard | 82.9 | - | 73.5 | - | 68.5 | 39.3 | 19.5 | 1.7 | 37.4 |
| | Parsel | 85.0 | +2.1 | - | - | - | - | - | - | - |
| | CodeT | 90.9 | +8.0 | 77.0 | +3.5 | 76.4 | 51.8 | 21.8 | 3.4 | 44.0 |
| | Reflexion | 91.0 | +8.1 | 77.1 | +3.6 | 71.3 | 41.1 | 19.5 | 2.5 | 38.6 |
| | MetaGPT | 85.9 | +3.0 | - | - | - | - | - | - | - |
| | FUNCODER | 94.5 | +11.6 | 79.5 | +6.0 | 83.1 | 58.0 | 26.4 | 3.4 | 48.6 |
| GPT-4o mini | Standard | 87.3 | - | 76.0 | - | 65.7 | 44.6 | 9.2 | 0.0 | 35.4 |
| | CodeT | 90.9 | +3.6 | 75.5 | -0.5 | 71.9 | 49.1 | 16.1 | 0.0 | 39.6 |
| | FUNCODER | 91.5 | +4.2 | 77.5 | +1.5 | 72.5 | 52.3 | 11.5 | 0.0 | 39.8 |
| Llama3$_{8b}$ | Standard | 61.6 | - | 60.5 | - | 9.0 | 1.8 | 0.0 | 0.0 | 3.6 |
| | CodeT | 68.9 | +7.3 | 61.5 | +1.0 | 12.4 | 0.0 | 0.0 | 0.0 | 4.4 |
| | FUNCODER | 79.7 | +18.1 | 62.5 | +2.0 | 22.0 | 0.9 | 0.0 | 0.0 | 8.0 |
| Codestral$_{22b}$ | Standard | 79.3 | - | 68.5 | - | 27.5 | 4.5 | 2.3 | 0.0 | 11.4 |
| | CodeT | 86.0 | +7.3 | 74.0 | +5.5 | 34.8 | 7.1 | 3.4 | 0.0 | 14.8 |
| | FUNCODER | 89.0 | +9.7 | 74.5 | +6.0 | 49.4 | 15.2 | 3.4 | 0.0 | 22.0 |
| StableCode$_{3b}$ | Standard | 61.0 | - | 51.5 | - | 7.3 | 0.9 | 0.0 | 0.0 | 2.8 |
| | CodeT | 75.0 | +14.0 | 57.5 | +6.0 | 11.2 | 1.8 | 0.0 | 0.0 | 4.6 |
| | FUNCODER | 81.0 | +20.0 | 63.5 | +12.0 | 13.5 | 4.5 | 1.1 | 0.0 | 6.2 |
| CodeLlama$_{34b}$ | Standard | 43.9 | - | 53.5 | - | 2.3 | 0.0 | 0.0 | 0.0 | 0.8 |
| | CodeT | 55.5 | +11.6 | 56.5 | +3.0 | 10.1 | 0.0 | 0.0 | 0.0 | 3.6 |
| | FUNCODER | 66.5 | +22.6 | 58.5 | +5.0 | 10.2 | 0.0 | 0.0 | 0.0 | 3.6 |
| StarCoder2$_{15b}$ | Standard | 59.8 | - | 64.5 | - | 18.0 | 0.9 | 2.3 | 0.0 | 7.2 |
| | CodeT | 70.7 | +10.9 | 66.0 | +1.5 | 13.5 | 0.9 | 0.0 | 0.0 | 5.0 |
| | FUNCODER | 78.7 | +18.9 | 70.0 | +5.5 | 29.2 | 4.5 | 0.0 | 0.0 | 11.6 |

Table 12: Experimental results on MATH, a competition-level mathematical reasoning benchmark. Best results are in bold. Text-based reasoning methods are denoted with $^\dagger$, while others use program-aided reasoning. We report both overall results and results in seven subjects: *Prealgebra*, *Algebra*, *Number Theory*, *Counting & Probability*, *Geometry*, *Intermediate Algebra*, and *Precalculus*.

| Model | Method | *Prealg.* | *Alg.* | *NT* | *Prob.* | *Geo.* | *InterAlg.* | *Precalc.* | **Overall** |
|---|---|---|---|---|---|---|---|---|---|
| GPT-3.5 | Standard$^\dagger$ | 62.2 | 37.4 | 20.0 | 29.8 | 31.0 | 24.4 | 21.8 | 34.6 |
| | CoT$^\dagger$ | 59.8 | 51.1 | 28.9 | 29.8 | 28.6 | 26.7 | 30.9 | 40.0 |
| | PoT | 68.3 | 50.4 | 33.3 | 48.9 | 21.4 | 18.2 | 29.1 | 41.0 |
| | Self-Refine | 74.4 | 49.6 | 48.9 | 57.4 | 28.6 | 35.6 | 36.4 | 48.6 |
| | FUNCODER | **76.8** | **61.2** | **55.6** | **59.6** | **34.1** | **36.0** | **41.8** | **54.0** |
| GPT-4 | Standard$^\dagger$ | 81.7 | 82.7 | 71.1 | 72.3 | **59.5** | 46.7 | 47.3 | 68.2 |
| | CoT$^\dagger$ | 84.1 | 87.1 | 62.2 | 68.1 | 45.2 | 48.9 | 54.5 | 68.6 |
| | PoT | 79.3 | 80.6 | 75.6 | 72.3 | 50.0 | 47.8 | 58.2 | 68.2 |
| | Self-Refine | 82.9 | 82.0 | 77.8 | 76.6 | 54.8 | 55.6 | **63.6** | 72.2 |
| | CR | 86.6 | 86.3 | **88.7** | 71.1 | 53.7 | 51.5 | 51.8 | 72.2 |
| | FUNCODER | **89.0** | **92.8** | 82.2 | **83.0** | 59.5 | **63.3** | 56.4 | **78.2** |
| GPT-4o mini | Standard$^\dagger$ | 79.3 | 83.5 | 75.6 | **87.2** | 47.6 | 57.8 | 56.4 | 71.8 |
| | CoT$^\dagger$ | **90.2** | **95.7** | **82.2** | 68.1 | 50.0 | **61.1** | **61.8** | **77.2** |
| | PoT | 80.5 | 84.2 | 77.8 | 72.3 | 50.0 | 60.0 | 50.9 | 71.0 |
| | Self-Refine | 79.3 | 83.5 | 75.6 | **87.2** | 47.6 | 57.8 | 56.4 | 71.8 |
| | FUNCODER | 81.7 | 83.5 | 80.0 | 80.9 | **59.5** | 60.0 | 54.5 | 73.2 |
| Llama3$_{8b}$ | CoT$^\dagger$ | 56.1 | **47.5** | 31.1 | 34.0 | **40.5** | 14.4 | **38.2** | 38.6 |
| | PoT | 67.1 | 32.4 | 24.4 | 34.0 | 16.7 | 21.1 | 18.2 | 32.6 |
| | FUNCODER | **67.9** | 45.7 | **51.1** | **53.2** | 19.0 | **37.8** | 30.9 | **45.0** |
| Codestral$_{22b}$ | PoT | 70.7 | 56.1 | **46.7** | 44.7 | 21.4 | 26.7 | 30.9 | 45.6 |
| | FUNCODER | **81.7** | **61.9** | **46.7** | **55.3** | **28.6** | **45.6** | **38.2** | **54.8** |
| StableCode$_{3b}$ | PoT | 20.7 | 14.4 | 17.8 | 25.5 | **4.8** | 8.9 | 9.1 | 14.4 |
| | FUNCODER | **46.3** | **30.2** | **20.0** | **29.8** | **4.8** | **20.0** | **18.2** | **26.6** |
| CodeLlama$_{34b}$ | PoT | 35.5 | 26.1 | 15.0 | 16.7 | 0.0 | 5.5 | 33.3 | 15.2 |
| | FUNCODER | **44.8** | **46.1** | **37.8** | **34.1** | **13.6** | **24.6** | **37.5** | **24.4** |
| StarCoder2$_{15b}$ | PoT | 46.3 | 29.5 | 28.9 | 25.5 | 21.4 | 27.8 | 23.6 | 30.2 |
| | FUNCODER | **72.0** | **39.6** | **40.9** | **46.8** | **23.8** | **28.1** | **27.3** | **40.8** |

Table 13: Full results of each method at different levels of MATH. The best results are in bold. Text-based reasoning methods are denoted with $^\dagger$, while others use program-aided reasoning.

| Model | Method | Level 1 | Level 2 | Level 3 | Level 4 | Level 5 | Overall |
|---|---|---|---|---|---|---|---|
| GPT-3.5 | Standard$^\dagger$ | 61.5 | 51.1 | 43.5 | 25.9 | 17.7 | 34.6 |
| | CoT$^\dagger$ | 76.9 | 48.9 | 50.9 | 33.6 | 21.8 | 40.0 |
| | PoT | 61.5 | 51.1 | 56.5 | 33.6 | 24.1 | 41.0 |
| | Self-Refine | **84.6** | 61.1 | 65.7 | 32.8 | 31.3 | 48.6 |
| | FUNCODER | **84.6** | **65.9** | **68.5** | **43.1** | **37.4** | **54.0** |
| GPT-4 | Standard$^\dagger$ | 89.7 | 85.6 | **83.3** | 55.2 | 51.0 | 68.2 |
| | CoT$^\dagger$ | 94.9 | 81.1 | 77.8 | 64.7 | 50.3 | 68.6 |
| | PoT | 94.9 | 80.0 | 74.1 | 63.8 | 53.1 | 68.2 |
| | Self-Refine | 94.9 | 81.1 | **83.3** | 62.1 | 60.5 | 72.2 |
| | CR | 90.7 | **90.0** | 81.9 | 66.4 | 52.2 | 72.2 |
| | FUNCODER | **94.9** | **90.0** | 81.5 | **75.9** | **66.0** | **78.2** |
| GPT-4o mini | Standard$^\dagger$ | 87.2 | 82.2 | 80.6 | 62.9 | 61.9 | 71.8 |
| | CoT$^\dagger$ | 97.4 | 90.0 | 87.0 | 71.6 | 61.2 | 77.2 |
| | PoT | 89.7 | 81.1 | 76.9 | 63.8 | 61.2 | 71.0 |
| | Self-Refine | 87.2 | 82.2 | 80.6 | 62.9 | 61.9 | 71.8 |
| | FUNCODER | 94.9 | 82.2 | 81.5 | 62.9 | 63.9 | 73.2 |
| Llama3$_{8b}$ | CoT$^\dagger$ | 76.9 | 46.7 | 46.3 | 25.9 | **27.9** | 38.6 |
| | PoT | 64.1 | 43.3 | 41.7 | 25.0 | 17.0 | 32.6 |
| | FUNCODER | **79.5** | **60.0** | **52.3** | **37.4** | **27.9** | **45.0** |
| Codestral$_{22b}$ | PoT | 79.5 | 56.7 | 57.4 | 34.5 | 29.9 | 45.6 |
| | FUNCODER | 84.6 | 66.7 | 67.6 | 43.1 | 39.5 | 54.8 |
| StableCode$_{3b}$ | PoT | 35.9 | 22.2 | 19.4 | 7.8 | 5.4 | 14.4 |
| | FUNCODER | **53.8** | **37.8** | **35.2** | **21.6** | **10.2** | **26.6** |
| CodeLlama$_{34b}$ | PoT | 36.1 | 30.7 | 28.0 | 13.0 | 8.8 | 15.2 |
| | FUNCODER | **60.6** | **52.1** | **44.3** | **28.8** | **16.3** | **24.4** |
| StarCoder2$_{15b}$ | PoT | 43.6 | 44.4 | 45.4 | 20.7 | 14.3 | 30.2 |
| | FUNCODER | 71.8 | 57.8 | 55.1 | 26.7 | 23.3 | 40.8 |

# B Examples

We provide example solutions for the baseline and FUNCODER on code generation and mathematical reasoning. All results are generated with GPT-3.5.

## B.1 Code Generation

**Info:** `xCodeEval/ff3c39b759a049580a6e96c66c904fdc` (binary search, difficulty 1800)

**Problem**
```
time limit per test: 1 second
memory limit per test: 256 megabytes
input: standard input
output: standard output

## Description

You have two friends. You want to present each of them several
    positive integers. You want to present cnt1 numbers to the first
    friend and cnt2 numbers to the second friend. Moreover, you want
    all presented numbers to be distinct, that also means that no
    number should be presented to both friends.In addition, the first
    friend does not like the numbers that are divisible without
    remainder by prime number x. The second one does not like the
    numbers that are divisible without remainder by prime number y. Of
     course, you're not going to present your friends numbers they don
    't like.Your task is to find such minimum number v, that you can
    form presents using numbers from a set 1,2,...,v. Of course you
    may choose not to present some numbers at all.A positive integer
    number greater than 1 is called prime if it has no positive
    divisors other than 1 and itself.

## Input

The only line contains four positive integers cnt1, cnt2, x, y (1<=
    cnt1,cnt2<109; cnt1+cnt2<=109; 2<=x<y<=3*104)- the numbers that
    are described in the statement. It is guaranteed that numbers x, y
     are prime.

## Output

Print a single integer - the answer to the problem.

## Sample Input 1

```
3 1 2 3
```

## Sample Output 1

```
5
```

## Sample Input 2

```
1 3 2 3
```

## Sample Output 2

```
```

```
4
```

## Notes

In the first sample you give the set of numbers {1,3,5} to the first
    friend and the set of numbers {2} to the second friend. Note that
    if you give set {1, 3, 5} to the first friend, then we cannot give
     any of the numbers 1, 3, 5 to the second friend. In the second
    sample you give the set of numbers {3} to the first friend, and
    the set of numbers {1, 2, 4} to the second friend. Thus, the
    answer to the problem is 4.

### B.1.1  Standard

```python
def main() -> None:
    cnt1, cnt2, x, y = list(map(int, input().split()))
    # binary search
    l, r = 1, 2 * (cnt1 + cnt2)  # upper bound
    while l < r:
        mid = (l + r) // 2
        # number of numbers that can be given to the first friend
        cnt_x = mid // x
        # number of numbers that can be given to the second friend
        cnt_y = mid // y
        # number of numbers that can be given to both friend
        cnt_xy = mid // (x * y)
        # number of numbers that can't be given to any friend
        cnt_none = mid - cnt_x - cnt_y + cnt_xy
        if cnt_x >= cnt1 and cnt_y >= cnt2 and cnt_none >= cnt1 + cnt2
    :
            r = mid
        else:
            l = mid + 1
    print(l)
```

**Result:** Wrong answer on sample test 1: expected: '5', found: '8' [1th token] ✗

### B.1.2  FUNCODER

```python
def main() -> None:
    cnt1, cnt2, x, y = map(int, input().split())

    left = 1
    right = int(2e9)
    while left < right:
        mid = (left + right) // 2
        if is_valid(mid, x, y, cnt1, cnt2):
            right = mid
        else:
            left = mid + 1

    print(left)

def is_valid(v, x, y, cnt1, cnt2):
    # check if there are cnt1 numbers not divisible by x
    non_x_numbers = v - (v // x)
    if non_x_numbers < cnt1:
        return False

    # check if there are cnt2 numbers not divisible by y
    non_y_numbers = v - (v // y)
    if non_y_numbers < cnt2:
        return False
```

```
        # check if there are cnt1+cnt2 numbers divisible by both x and y
        lcm = (x * y) // gcd(x, y)
        common_numbers = v - (v // lcm)
        if common_numbers < cnt1 + cnt2:
            return False

        return True

def gcd(a: int, b: int) -> int:
    """get the greatest common divisor of two numbers $a$ and $b$"""

    while b:
        a, b = b, a % b
    return a
```

**Result:** Accepted ✓

## B.2  Mathematical Reasoning

**Info:** `MATH/number_theory/42` (level 5)

**Problem**  What is the sum of the smallest and second-smallest positive integers $a$ satisfying the congruence $27a \equiv 17 \pmod{40}$ ?

**Ground Truth:** 62

### B.2.1  CoT

```
We can solve for $a$ by multiplying both sides of the congruence by
    the modular inverse of $27$ modulo $40$. Since $27$ and $40$ are
    relatively prime, we know that $27$ has a modular inverse modulo
    $40$. We can find this inverse using the Euclidean algorithm:
\begin{align*}
\text{gcd}(40,27) &= \text{gcd}(27,13)
&= \text{gcd}(13,1)
&= 1.
\end{align*}
Therefore, the modular inverse of $27$ modulo $40$ exists. We can find
    it using the Euclidean algorithm backwards:
\begin{align*}
1 &= 13 - 3 \cdot 4
&= 13 - 3(40 - 27)
&= 4 \cdot 27 - 3 \cdot 40.
\end{align*}
Thus, the modular inverse of $27$ modulo $40$ is $4$. Multiplying both
    sides of the congruence $27a\equiv 17 \pmod{40}$ by $4$ gives
    $108a\equiv 68 \pmod{40}$. Simplifying the left-hand side gives
    $8a\equiv 28 \pmod{40}$. Dividing both sides by $4$ gives $2a\
    equiv 7 \pmod{10}$. The smallest positive integer $a$ satisfying
    this congruence is $a=9$, and the second-smallest is $a=19$.
    Therefore, the sum of the smallest and second-smallest positive
    integers $a$ satisfying the congruence $27a\equiv 17 \pmod{40}$ is
    $9+19=\boxed{28}$.
```

**Result:** 28 ✗

### B.2.2 PoT

```python
def solution():
    r"""What is the sum of the smallest and second-smallest positive
    integers $a$ satisfying the congruence $$27a\equiv 17 \pmod{40}~?
    $$"""
    k = 1
    candidates = []
    while len(candidates) < 2:
        a = (40 * k + 17) // 81
        if a > 0 and (40 * k + 17) % 81 == 0:
            candidates.append(a)
        k += 1

    return sum(candidates)
```

**Result:** 74 ✗

### B.2.3 FUNCODER

```python
from typing import Tuple

def solution():
    r"""What is the sum of the smallest and second-smallest positive
    integers $a$ satisfying the congruence $$27a\equiv 17 \pmod{40}~?
    $$"""

    a = 27
    m = 40
    b = 17
    inverse = find_inverse(a, m)
    x = (b * inverse) % m

    # Find the smallest and second-smallest positive integers
    satisfying the congruence equation
    # using the formula: x + k * (m // gcd(a, m))
    gcd, _, _ = extended_euclid(a, m)
    s1 = x
    s2 = x + (m // gcd)
    return s1 + s2

def find_inverse(a: int, m: int) -> int:
    """find the inverse of $a$ modulo $m$"""

    _, x, _ = extended_euclid(a, m)
    return x % m if x >= 0 else (x + m) % m

def extended_euclid(a: int, b: int) -> Tuple[int, int, int]:
    """Extended Euclidean Algorithm"""
    if b == 0:
        return a, 1, 0
    gcd, x1, y1 = extended_euclid(b, a % b)
    x = y1
    y = x1 - (a // b) * y1
    return gcd, x, y
```

**Result:** 62 ✓

# C Prompts

## C.1 Prompt for Standard

```
You are a programming copilot, you can solve a problem by writing
    Python functions. Your task is to:

  - You need to write a Python function that returns the answer.
  - You can import libraries to better solve the problem.
  - Do not write any code outside the function (importing is accepted)
    .
```

```
<User>:

Let's think step by step and complete the following Python function `
    sum_factor` that solves:

```python
def sum_factor(a: int, b: int) -> int:
    """Return the sum of all common prime factors of $a$ and $b$"""

    raise NotImplementedError()
```
```

```
<Assistant>:

First, I need to get the prime factors of $a$ and $b$.
Second, I can use `for` loop to find common element in two factors
    list.
Here is the `sum_factor` function:

```python
def sum_factor(a: int, b: int) -> int:
    """Return the sum of all common prime factors of $a$ and $b$"""
    factors_a = prime_factor(a)
    factors_b = prime_factor(b)
    common_factors = get_common(factors_a, factors_b)
    return sum(common_factors)

def prime_factor(x: int) -> list:
    """get a list of prime factors of number $x$"""
    ret = []
    i = 1
    while i * i <= x:
        i += 1
        if x % i == 0 and is_prime(i):
            ret.append(i)
    return ret

def is_prime(x: int) -> bool:
    """determine $x$ is a prime number or not"""
    if x < 2:
        return False
    for i in range(2, int(x**0.5) + 1):
        if x % i == 0:
            return False
    return True

def get_common(a: list, b: list) -> list:
    """get common element in two list $a$ and $b$"""
    ret = []
    for item in a:
        if item in b:
            ret.append(item)
    return ret
```

```
```

<User>:

Let's think step by step and complete the following Python function `{
    cur_func_name}` that solves:

```python
{cur_func}
```
```

## C.2 Prompt for Divide

```
You are a programming copilot, you can solve a problem by writing
    Python functions. Your task is to:

  - For every turn, you need to write a Python function that returns
    the answer based on Current Code (not code in chat history).
  - Do not modify function name, arg names, docstring in given
    functions.
  - You can import libraries to better solve the problem.
  - If a single function is too hard to solve, you can decompose it
    into multiple smaller functions.
  - You can leave new function unimplemented for now, but write the
    function at the end of the code and comment what the function does
    ."
```

```
<User>:

Current Code:
```python
def sum_common_factors(a: int, b: int) -> int:
    """Compute the sum of all common prime factors of $a$ and $b$"""
    raise NotImplementedError()
```

Let's think step by step and complete the following Python function `
    sum_common_factors` that solves:
"Compute the sum of all common prime factors of $a$ and $b$"
```

```
<Assistant>:

First, I need to get the prime factors of $a$ and $b$.
Second, I can use `for` loop to find common element in two factors
    list.
Finally, sum the common factor list and return the answer.
Here is the `sum_common_factors` function:

```python
def sum_common_factors(a: int, b: int) -> int:
    """Compute the sum of all common prime factors of $a$ and $b$"""
    factors_a = prime_factor(a)
    factors_b = prime_factor(b)
    common_factors = get_common(factors_a, factors_b)
    return sum(common_factors)

def prime_factor(x: int) -> list:
    """get a list of prime factors of number $x$"""
    raise NotImplementedError()

def get_common(a: list, b: list) -> list:
    """get common element in two list $a$ and $b$"""
    raise NotImplementedError()
```
```

```
<User>:

Current Code:
```python
def sum_common_factors(a: int, b: int) -> int:
    """Compute the sum of all common prime factors of $a$ and $b$"""
    factors_a = prime_factor(a)
    factors_b = prime_factor(b)
    common_factors = get_common(factors_a, factors_b)
    return sum(common_factors)

def get_common(a: list, b: list) -> list:
    """get common element in two list $a$ and $b$"""
    raise NotImplementedError()
```

Let's think step by step and complete the following Python function `
    get_common` that solves:
"get common element in two list $a$ and $b$"

<Assistant>:

Here is the `get_common` function:

```python
def get_common(a: list, b: list) -> list:
    """get common element in two list $a$ and $b$"""
    ret = []
    for item in a:
        if item in b:
            ret.append(item)
    return ret
```

<User>:

Current Code:
```python
{prev_code}
```

Let's think step by step and complete the following Python function `{
    cur_func_name}` that solves:
"{cur_func_doc}"
```

## C.3    Prompt for Conquer

```
You are a programming copilot, you can solve a problem by writing
    Python functions. Your task is to:

  - For every turn, you need to write a Python function that returns
    the answer, based on current code (not code in chat history) and
    problem description.
  - Do not modify function name, arg names, docstring in given
    functions.
  - Consider reusing existing functions that are already implemented.
  - You can import libraries to better solve the problem.

<User>:

Current Code:

```python
def prime_factor(x: int) -> list:
    """get a list of prime factors of number $x$"""
```

35
```

```python
    ret = []
    i = 1
    while i * i <= x:
        i += 1
        if x % i == 0 and is_prime(i):
            ret.append(i)
    return ret

def is_prime(x: int) -> bool:
    """determine $x$ is a prime number or not"""
    if x < 2:
        return False
    for i in range(2, int(x**0.5) + 1):
        if x % i == 0:
            return False
    return True

def get_common(a: list, b: list) -> list:
    """get common element in two list $a$ and $b$"""
    ret = []
    for item in a:
        if item in b:
            ret.append(item)
    return ret

def sum_common_factors(a: int, b: int) -> int:
    """Return the sum of all common prime factors of $a$ and $b$"""

    raise NotImplementedError()
```

Let's think step by step and implement the following method `sum_common_factors` using existing functions to solve:
"Return the sum of all common prime factors of $a$ and $b$"

<Assistant>:

First, I need to get the prime factors of $a$ and $b$.
Second, I can use `for` loop to find common element in two factors list.
Finally, sum the common factor list and return the answer.
Here is the `sum_common_factors` function:

```python
def sum_common_factors(a: int, b: int) -> int:
    """Compute the sum of all common prime factors of $a$ and $b$"""
    factors_a = prime_factor(a)
    factors_b = prime_factor(b)
    common_factors = get_common(factors_a, factors_b)
    return sum(common_factors)
```

<User>:

Current Code:

```python
{prev_code}
```

Let's think step by step and implement the following method `{cur_func_name}` using existing functions to solve:
"{cur_func_doc}"

## C.4 Prompt for Generate Possible Input

```
You are a proficient software engineer and architect, specialized in
    testing, capable of observing and finding security holes and
    logical issues in code that spans across multiple computer science
     fields and mathematics. You are given a piece of Python function,
     and are expected to produce some function calls for that function
    . Specifically:

  - You should invoke the function in a one-liner fashion.
  - Do not bring in imports other than what's already imported. Use
    the pre-declared imports in the original function only.
  - The callee may have multiple arguments, treat them with care.
  - You **must** respect the function signature and docstring, and be
    aware so you don't generate illegal inputs.
  - Keep the inputs simple but general, and that either edge cases or
    common cases are meaningful.

Here follows a series of mutually uncorrelated functions to test, one
    per conversation.

<User>:

Let's think step by step and create some tests for the following
    function `check_valid_brackets(...)` in Python.

```python
def check_valid_brackets(seq: str) -> bool:
    """Determine if a bracket sequence consisting of '(', ')', '{',
    '}', '['
    and ']' is valid."""

    mapping = {')': '(', '}': '{', ']': '['}
    stack = []
    for c in seq:
        if c in mapping:
            if not stack or stack[-1] != mapping[c]:
                return False
            stack.pop()
        else:
            stack.append(c)
    return not stack
```

Store your function calls for `check_valid_brackets(...)` as function
    callss, one per line. They will be called later.

<Assistant>:

Sure, I can create some function calls for the `check_valid_brackets`
    function. We can either choose to test it with a valid bracket
    sequence or an invalid one. Empty strings are also considerable.
    Here are some function calls for the function:

```python
check_valid_brackets("()")  # True
check_valid_brackets("(([[]])")  # True
check_valid_brackets("((())")  # False
check_valid_brackets("()[]{}")  # True
check_valid_brackets("([)]")  # False
check_valid_brackets("")  # True
check_valid_brackets(")(")  # False
```

<User>:
```

```
Let's think step by step and create some tests for the following
    function `{cur_func_name}(...)` in Python.

```python
{prev_code}
```

Store your function calls for `{cur_func_name}(...)` as function
    callss, one per line. They will be called later.
```

## C.5 Prompt for Self-Test Generation

```
You are a proficient software engineer and architect, specialized in
    testing, capable of observing and finding security holes and
    logical issues in code that spans across multiple computer science
     fields and mathematics. You are given a piece of Python function,
     and are expected to produce some test cases for that function.
    Specifically:

  - You should invoke the function and assert its results in a one-
    liner fashion.
  - Do not bring in imports other than what's already imported. Use
    the pre-declared imports in the original function only.
  - The callee may have multiple arguments, treat them with care.
  - You **must** respect the function signature and docstring, and be
    aware so you don't generate illegal inputs.
  - Keep the inputs & outputs simple but general, and that either edge
     cases or common cases are meaningful.

Here follows a series of mutually uncorrelated functions to test, one
    per conversation.

<User>:

Let's think step by step and create some tests for the following
    function `lcm(...)` in Python.

```python
def lcm(a: int, b: int) -> int:
    """Find the least common multiple of `a` and `b`. Samples:

    >>> lcm(3, 5)
    15
    >>> lcm(4, 6)
    12
    """

    return round(a * b / gcd(a, b))
```

Store your test cases for `lcm(...)` as assertions, one per line. They
     will be called later.

<Assistant>:

Sure, I can create some test cases for the `check_valid_brackets`
    function. We consider the following cases: 1. the two operands are
     not co-prime and has common factors; 2. the two operands are
    equal; 3. one of them is 1; 4. two of them is 1; 5. both operands
    are primes. Here is an example of these test cases in Python:

```python
assert lcm(15, 25) == 75
assert lcm(32, 32) == 32
```
```

```
assert lcm(1, 5) == 5
assert lcm(1, 1) == 1
assert lcm(17, 19) == 17 * 19
```
<User>:

Extract tests for the following function `{cur_func_name}(...)` in
    Python.

```python
{prev_code}
```

Store your test cases for `{cur_func_name}(...)` as assertions, one
    per line. They will be called later.
```

## C.6 Prompt for MATH judging

```
You are a mathematical teacher, your task is to:

    - Judge whether the prediction is matching the answer
    - Output "Judge: Correct." or "Judge: Wrong.", please do not
    output redundant words
    - Numerical errors should be ignored ($1$ is equal to $0.99999998$
    )
    - Some answer might be represent in latex format, and some might
    be float number, this should be consider as correct ($\frac{1}{2}$
     is equal to $0.5$, $3$ $\sqrt{66}$ is equal to $24.37211$)
    - Unit in answer should be ignored, and should be consider as
    correct ($13 cm^2$ is equal to $13.0$, $\$13$ is equal to $13$)

Now, the answer and prediction is:
Answer: {ground_truth}
Prediction: {model_output}
Please output "Judge: Correct." if two answers are literally the same,
    or "Judge: Wrong." for not same, please do not output redundant
    words.
```

