# OpenReview forum: "Divide-and-Conquer Meets Consensus: Unleashing the Power of Functions in Code Generation"
_NeurIPS.cc/2024/Conference — NeurIPS 2024 oral_

### Official Review · Reviewer_s7Dq · 2024-06-17

**Soundness:** 2
**Presentation:** 2
**Contribution:** 2
**Rating:** 5
**Confidence:** 4

**Summary:**

The paper proposes FunCoder, a code generation framework incorporating the divide-and-conquer strategy with functional consensus. FunCoder recursively branches off sub-functions as smaller goals during code generation, represented by a tree hierarchy. These sub-functions are then composited to attain more complex objectives. Additionally, they use a consensus formed by identifying similarities in program behavior, mitigating error propagation. FunCoder shows great perormance compared to baselines by +9.8% on average in HumanEval, MBPP, xCodeEval and MATH with GPT-3.5 and GPT-4.

**Strengths:**

+ interesting method
+ impressive experiment and results

**Weaknesses:**

- some important details are missing
- lack analysis for cost

The paper is commendably clear and the results of the experiments are striking. However, I find that the sections detailing the approach are somewhat brief, which leaves me wanting a deeper understanding of the full method, especially since the study focuses on employing a divide-and-conquer approach for code generation. It would be greatly beneficial if the authors could expand on this strategy. For instance, how does the model determine which functions require further division or are enough simplistic? Are there specific experiments that illustrate this process? Additionally, it would be insightful to know whether the model is designed to recognize dependencies among the subdivided functions and how it integrates these components effectively.

Regarding the implementation costs, the approach involves segmenting the program generation into multiple stages using a depth-first search for each sub-program. This intricate process prompts a question about the overall efficiency and resource utilization. I would appreciate if the authors could provide more detailed information on the costs associated with these procedures.

**Questions:**

- How the model decide the functions to be further divided or too simple? Any experiment?
- How the conquer works? Does the model aware of the dependency among those sub functions?
- What is the cost of each part?

**Limitations:**

Some limitations have been discussed but it would be better to discuss more on the cost of the FUNCODER and the limitation of divide-and-conquer for hard problems.

---

> ### Author Rebuttal · Authors · 2024-08-06
>
> Thank you for your time and effort in reviewing our paper! These are very helpful suggestions and we address your questions here:
>
> > **W1: The sections detailing the approach are somewhat brief, which leaves me wanting a deeper understanding of the full method, especially since the study focuses on employing a divide-and-conquer approach for code generation.**
>
> Apologies for any confusion. Due to page constraints, we couldn't describe the complete divide-and-conquer process in detail in the main text. However, we supplemented the Implementation Details in Appendix A.1 and have respective prompts in Appendix C to help understand the model's operation. As described in Algorithm 1, FunCoder is a recursive process following a DFS pattern. We use square brackets [L1] below to denote the line number in Algorithm 1.
>
> - FunCoder [L1], when solving each function $f$, first performs the *Divide* stage [L3-L9], where the LLM initially writes the function and identifies some potential sub-problems, represented as sub-function stubs (e.g., `def f(xs: list[int]) -> int`) [L3]. In this process, we identify the sub-problems of the current problem, thereby understanding the dependency relationship between functions.
> - For each decomposed sub-problem $g_1, g_2, \ldots$, we recursively use FunCoder to obtain the final implementation $G_i$ for that sub-problem [L5-L8]. This $G_i$ shall replace the previously incomplete subfunction stub signature in the final program.
>   - FunCoder [L1], when solving each function $g_1$, …
>   - FunCoder [L1], when solving each function $g_2$, …
> - Now that all sub-problems of $f$ are implemented, we move on to the *Conquer* stage [L11-13] to complete the larger problem. By combining the signature $f$ and the final implementations $G_i$ of sub-problems, we generate the complete implementation $F$ [L11] and return it [L13].
>
> Let's describe how this algorithm works in detail by combining it with the example given in the lower half of Figure 1.
>
> ```py
> [a.1] FunCoder(a)
> │  [a.3] LLM(a) -> A, {b, c}       # divide
> ├──[b.1] FunCoder(b)
> │  │  [b.3] LLM(b) -> B, {d, e}    # divide
> │  ├──[d.1] FunCoder(d)
> │  │  │  [d.3] LLM(d) -> D, {}     # divide
> │  │  └──[d.13] return D           # nothing to conquer
> │  ├──[e.1] FunCoder(e)
> │  │  │  [e.3] LLM(e) -> E, {}     # divide
> │  │  └──[e.13] return E           # nothing to conquer
> │  │  [b.11] LLM(B, {D, E}) -> B*  # conquer
> │  └──[b.13] return B*
> ├──[c.1] FunCoder(c)
> │  │  [c.3] LLM(c) -> C, {}        # divide
> │  └──[c.13] return C              # nothing to conquer
> │  [a.11] LLM(A, {B, C}) -> A*     # conquer
> └──[a.13] return A*                # final result
> ```
>
> We will add this content to the Appendix to provide a more detailed explanation of our method. Hope this addresses your concerns about our approach.
>
> > **Q1: How the model decide the functions to be further divided or too simple? Any experiment?**
>
> We rely on the knowledge of code and instruction following capabilities acquired by the LLM during its training. This enables the LLM to make approximate decisions on whether a function should be further decomposed. As stated in Appendix A.1, the LLM is prompted in the divide phase to implement the current function and simultaneously introduce new sub-functions if necessary.
>
> As shown in Appendix C.2, we designed the *Divide* prompt with a 2-shot example. One example demonstrates that complex problems should be decomposed, while the other example shows that simple problems can be implemented without new sub-functions.
>
> Furthermore, in Table 5, we observed that functions produced in the *Divide* stages are highly domain-specific. These sub-functions are likely to be generated based on knowledge from pre-training data, supporting how LLM can decompose tasks based on pretrained knowledge.
>
> > **Q2: How the conquer works? Does the model aware of the dependency among those sub functions?**
>
> Regarding the complete process of divide-and-conquer, why Conquer works, and whether the model is aware of function dependencies, we have briefly discussed this in response to Weakness 1. Here, we elaborate further:
>
> During the *Divide* phase, we extract the hierarchical relationships between functions to construct a dependency tree. *Conquer* merges finished sub-problems, and this is entirely consistent with the principles of divide-and-conquer. In our task, *Conquer* provides the finalized sub-functions in the context and rewrites the parent function based on them. We will make this point clear in future versions.
>
> Conquer is indispensable. In the *Divide* phase, the function is generated before the sub-problem definitions, so the details of the sub-problems are not clear at this point. And the invocation of sub-functions or positions from where they are invoked by the parent function might be incorrect. It is therefore necessary to regenerate the function after all sub-functions are finished. In Table 3, we conducted an ablation study on the *Conquer* stage (Two-pass vs. One-pass) and empirically found that enabling Conquer significantly improves the code generation performance.
> > **Q3: What is the cost of each part? What about overall efficiency and resource utilization.**
>
> In Appendix A.6, Table 11, we listed the token usage of gpt-3.5-turbo for both our method and the baselines. We have also provided further statistics on token usage in our General Response. Upon your request, we have further broken down the token usage for each stage of FunCoder in the following table. Hope this answers your questions.
>
> | Part in FunCoder | mean tks | med tks |
> | --- | --- | --- |
> | TOTAL | 5402.0 | 5166.5 |
> | Divide | 1241.0 | 1135.0 |
> | Input generation | 930.9 | 861.5 |
> | Conquer | 3230.2 | 3149.5 |
>
> Our algorithm itself is very lightweight and does not consume much CPU or RAM resources. Resource consumption comes from calling the LLMs. For an overall analysis of token efficiency, please refer to the general response. We will include these in our next revision.

---

> > ### Comment · Reviewer_s7Dq · 2024-08-09
> >
> > Thank you for the detailed rebuttal and the efforts made to clarify the cost analysis in your paper. However, I still have some concerns regarding the logic and assumptions underlying this analysis.
> >
> > The analysis of token cost appears to assume an ideal scenario where no failures occur. Additionally, the new data presented in the rebuttal seems to mirror Table 3 of the paper(ablation study). We would like to see the analysis results of Table 1 or 2(compared to the SOTA).
> >
> > In a naive program, the standard is a -> BC, and yours is A, [B, C] (follow your notations).
> > We assume the divide and conquer is perfectly gained, and the pass rate is just the success rate.
> > The best performance of FUNCODER is 94.5.
> > Then, your success rate to derive b->B and c->C should be increased to the square root of 0.945, which means 0.972.
> >
> > When the program becomes much more complex, the requirement of success rate for each divided subprogram is further increased.
> >
> > I think there should be some design and cost to increase the performance, even if the divided programs are simpler than the original ones.
> > However, from the rebuttal, it seems most parts depend on the LLM for program generation and divide-conquer, which means the error will increase when the program becomes more complex.
> >
> > I am willing to consider increasing my score if we could address these questions.

---

> > > ### Author Response · Authors · 2024-08-11
> > >
> > > We thank you again for taking your time to read our responses and provide suggestions.
> > >
> > > > **1. The analysis of token cost appears to assume an ideal scenario where no failures occur.**
> > >
> > > Indeed, in the analysis of general response, we simplified the situations and ignored the the situations where the LLMs make a mistake. However, in token cost results (Table 3, 11 and in our response) we have **always included** token usage for retries to ensure fair comparison, and we argue that this situation may be trivial to consider.
> > >
> > > Failures that cause retries can rarely happen in FunCoder -- they happen if and only if no code can be extracted from the responses' Markdown blocks, or that the code was syntactically incorrect. We provide additional statistic results in the table below, which shows a relatively low failure rate even in complex questions (MATH or xCodeEval).
> > >
> > > | Dataset | Pass@1 | failed to parse / all LLM calls |
> > > |---|---|---|
> > > | HumanEval | 85.4 | 0.10% |
> > > | MBPP | 78.5 | 0.31% |
> > > | xCodeEval | 31.4 | 1.68% |
> > > | MATH | 54.0 | 2.10% |
> > >
> > > > **2. We would like to see the analysis results of Table 1 or 2 (compared to the SOTA).**
> > >
> > > We report token consumption data for some of the fields in Table 1 and Table 4, regarding GPT-3.5 and SOTA methods, in the table below. Some of the cells are missing data since some methods were not open-source and did not report detailed token usage.
> > >
> > > | Dataset | Method | Pass@1 | min tks | max tks | mean tks | med tks |
> > > |---|---|---|---|---|---|---|
> > > | HumanEval | Standard | 68.3 | 648 | 1477 | 886.7 | 861 |
> > > | | CodeT | 81.1 | 2298 | 9645 | 4479.1 | 4166 |
> > > | | Reflexion | 69.5 | 416 | 4906 | 1416.1 | 754 |
> > > | | LDB (Reported prev SOTA) | 82.9 | - | - | ~23000 | - |
> > > | | FUNCODER | 85.4 | 3015 | 13850 | 5402.0 | 5166 |
> > > | xCodeEval | Standard | 20.2 | 1051 | 3343 | 1599.5 | 1530 |
> > > | | CodeT (prev SOTA) | 23.2 | 2264 | 9245 | 3937.4 | 3733 |
> > > | | Reflexion | 20.6 | 2977 | 1003222 | 401767.3 | 328591 |
> > > | | FUNCODER | 31.4 | 4883 | 53225 | 10559.7 | 8927 |
> > > | MATH | Standard | 62.2 | 551 | 5867 | 953.0 | 835 |
> > > | | FUNCODER | 76.8 | 2622 | 30139 | 7075.5 | 5666.5 |
> > >
> > > > **3. Then, your success rate to derive b->B and c->C should be increased to the square root of 0.945, which means 0.972. When the program becomes much more complex, the requirement of success rate for each divided subprogram is further increased.**
> > >
> > > **Accumulated errors in FunCoder also occur in Standard**, where generating all functions (ABC) all at once would have a higher error rate than generating one at a time (A, B, C).
> > >
> > > Note that this metaphor may not be exact since 94.5% acc is an average of all problems, where different problems are decomposed into different depths. However, we follow this metaphor and continue to show why FunCoder works better than other methods. Here success rate of *Standard* on a single function is $\sqrt{82.9}=91.0$. On a harder dataset where problems are decomposed to 10 levels of functions on average, *Standard*'s overall accuracy will be $91.0^{10}=38.9$, while FunCoder still keeps $97.2^{10}=79.3$.
> > >
> > > Results on xCodeEval dataset shows that FunCoder gets greater improvements on harder problems compared to simple problems. But of course, the analysis we just made was based on a simple assumption and does not reflect reality.
> > >
> > > > **4. I think there should be some design and cost to increase the performance, even if the divided programs are simpler than the original ones.**
> > >
> > > We designed FunCoder under the consideration of complex problem performance, and have made visible progress. Particularly, our method:
> > >
> > >  1. **Divide** dynamically decomposes problems into simpler subproblems, making them easier to solve. The Divide stage always considers just one layer of problem at a time, thus can keep the model in context, reducing ambiguity and complexity caused by extra-long code. For instance, the problem A-B-C-D requires all these functions to be in context in *Standard*, while ours only need 2 for each of the 4 calls.
> > >  2. **Conquer** enables bottom-up generation through re-composing simple functions into a complex solution to a complex problem. This mitigates the issue where Divide cannot see the subfunctions while generating parent functions, making function dependencies more robust. (Note that LLMs are autoregressive models, so while *Standard* generates ABC, when A is being generated B,C aren't there yet. This could cause *Standard* to mis-invoke B or C from A in this implementation.)
> > >  3. **Functional consensus** verifies every function starting from the leaves. Through sampling and voting, incidental errors and cascading errors may be reduced and thus the accuracy could be improved. Before our method, self-test was widely used for verifying programs, and now FunCoder manages to achieve higher performance with the same level of token complexity.
> > >
> > > We further note that Divide, Conquer and Consensus may be used together, and that using them in conjunction will further raise the accuracy on complex problems.

---

> > > > ### Comment · Reviewer_s7Dq · 2024-08-11
> > > >
> > > > Thank you for your detailed response.
> > > > I am generally satisfied with the clarifications provided; however, I have reservations regarding point 3.
> > > > Traditional LLMs just generate the whole code.
> > > > Your techniques are generated in a divide-conquer way, which means the LLM needs to understand the code dependency and generate multiple times for each sub-problem considering the context.
> > > > The sub-problems may be easier than the origin problems, which means the overall success rate (94.5) and the success rate for sub-problems (97.2) would be different.
> > > > More complex problems require higher accuracy on sub-problems (~100%).
> > > > Does that mean that in your experiment, for simple sub-problems, the LLM can reach almost 100% accuracy?
> > > >
> > > > Additional experiments could be highly beneficial to solidify the claims of your technique’s effectiveness. Specifically, experiments that vary the depth of sub-problem generation (e.g., restricting the LLM to generate a depth of 2 or 3 programs) could provide valuable insights into how the structure and depth of problem decomposition affect the overall pass rate.
> > > >
> > > > I've adjusted my score to reflect my appreciation of the responses you have provided.

---

> > > > > ### Author Response · Authors · 2024-08-12
> > > > >
> > > > > We sincerely thank you for the follow-up discussion and the rating update.
> > > > >
> > > > > For accuracy on simple sub-problems, we hypothesize that well-trained LLMs can solve them well (compared to complex sub-problems). But since this needs human labeling for all LLM outputs, which is beyond our available man-power, we might not include statistics for this issue. With that said, we hope that future research may look into this and investigate further.
> > > > >
> > > > > While there may be a slight possibility of losing accuracy forcing the LLM to decompose a simple problem (when the LLM is not trained to decompose such problems), or vice versa, conducting experiments that enforce program depth during generation is actually a very interesting idea. Such experiments may provide more insights to how complexity correlates with functional decomposition, and may lead to more promising ways to reliable code generation. So we welcome future work that proposes new, faithful methods to control this function decomposition depth.

---

### Official Review · Reviewer_Ar6t · 2024-07-02

**Soundness:** 2
**Presentation:** 3
**Contribution:** 3
**Rating:** 6
**Confidence:** 4

**Summary:**

This paper presents Divide-and-Conquer Meets Consensus -- a prompting scheme for generating complex functional code. In contrast to planning ahead of time, the proposed technique performs planning in smaller steps by decomposing a coding task into smaller sub-tasks recursively and solving them when they are simple enough. The evaluation shows that the technique can significantly outperform prior arts and works for both proprietary models and smaller open models.

**Strengths:**

1. This paper targets the important problem of code generation for complex coding tasks which is right now the bottleneck of the code generation domain as simple coding benchmarks have been saturated.
2. The overall framework of the technique is novel and beautiful, borrowing insights from the classical concept of the divide-and-conquer principle.
3. This paper provides a number of interesting insights beyond the base experimental results, e.g., in self-testing it is challenging to obtain both valid code and valid tests.

**Weaknesses:**

1. There is a flaw in the design of functionality similarity. Authors claim in L104 that they sample inputs from the input domain of the function, namely D(f). First, it is non-trivial to infer D(f). While simple type analysis is applicable, oftentimes the pre-conditions of the function can go beyond type constraints. For example, the pre-condition asserts the input sequence is sorted when doing code generation for binary search. Second, it is challenging to accurately infer such pre-conditions, and inputs violating such pre-conditions often manifest undefined behaviors of function candidates, falsifying functionality-similarly candidates. For example, in the EvalPlus paper, these conditions are manually specified to ensure the soundness of test input generation. In summary, the equivalence-modulo-inputs idea requires knowing the accurate input domain and designing corresponding input samplers, which are oversimplified in this paper.
2. Back to the high-level framework, it is unclear why functionality consensus would work in the first place. Functionality consensus selects the most common functor candidate, but commonality may not necessarily correlate with correctness in code generation. Correct samples can even be exceptional in sampling when the task is challenging to the LLM.
3. The diversity of studied models is a bit limited. It would be helpful to run Divide-Conquer on more open models such as StarCoder2.

Minor suggestions:
1. L97 "A program specifies its functionality (or behavior) through the control flow...": This is not accurate as control flow is just a partial representation of code semantics. For example, `a + b` and `a - b` share the same control flow (i.e., no control flow) but stand for completely different semantics.

**Questions:**

1. Can you exemplify "cascading errors" in Section 2.2? Can you also better explain why sampling multiple functions helps mitigate such errors?
2. The technique is motivated to solve complicated coding tasks. I wonder if the selection of benchmarks can really exercise the core part of the technique, i.e., resolving complex code generation tasks.
3. When computing the token usage, did you include tokens of unused samples when computing "functionality similarity"? Can you also provide more statistics such as the medium token consumption in case the average number is biased by simple tasks? Overall the presented token consumption in Table 3 is much smaller than I expected according to the large prompts shown in the Appendix (and these will be called multiple times).
4. How's the token usage compared to other baselines?
5. Is the technique single-turn or multi-turn when being implemented for evaluation?

**Limitations:**

Social impact wise this paper is fine. For technical limitations, they are reflected in the weaknesses section.

---

> ### Author Rebuttal · Authors · 2024-08-06
>
> We thank you for your time and effort in reviewing our paper! We find your suggestions very helpful and we hereby address your questions:
>
> > **W1: The equivalence-modulo-inputs idea requires knowing the accurate input domain and designing corresponding input samplers, which are oversimplified in this paper.**
>
> It is indeed challenging to *ensure* that LLMs generate reliable and valid inputs. However, in order to verify code correctness using LLMs, there are typically only three major methods:
>
> 1. Predict bugs by just reading the code (e.g. self-refine).
> 2. Generate unit tests for self-testing (e.g. CodeT).
> 3. (Ours) Generate inputs and select the best program based on outputs of sampled programs.
>
> In our comparisons, we focused more on the widely used unit-test method, which not only has to generate reliable inputs but also provide correct case outputs. If the generated output is incorrect, program verdicts will be deeply impacted. Experiment results in Table 3 empirically show a clear advantage of our method over unit tests.
>
> > **W2: Functionality consensus selects the most common functor candidate, but commonality may not necessarily correlate with correctness in code generation. Correct samples can even be exceptional in sampling when the task is challenging to the LLM.**
>
> Improving performance beyond pre-trained capabilities is hard. While it's quite challenging to prove it formally, our consensus still works better than clustering and other methods, as is shown in Table 3. We look forward to future work substantiating this finding.
>
> Admittedly, commonality is less likely to boost performance on problems beyond the LLMs' knowledge, but it isn't necessarily outperformed by the *Standard* baseline. However, commonality empirically reduces incidental errors, as we exemplified in general response.
>
> Refer to xCodeEval results in Table 1. Although our method yields little improvements on Expert-level problems where LLMs can almost never get right, it happened to perform quite well on Hard-level and easier problems. This can be similarly observed on MATH dataset.
>
> > **W3: The diversity of studied models is a bit limited. It would be helpful to run on more open models such as StarCoder2.**
>
> Thanks for your suggestion. We adapted our approach and found that FunCoder can also achieve good results with StarCoder2-15b, bringing its pass@1 of 59.8 on *Standard* to 78.7 with our method. More promising results can be found in our General Response.
>
> > **W4:  "A program specifies its functionality (or behavior) through the control flow..." is not accurate.**
>
> Thank you for pointing out this mistake. Indeed, considering only the control flow of a program is incomplete when it comes to behavior. This was an oversight during our writing process. We'll correct this in our next revision as "the program's control flow and logic".
>
> > **Q1: Can you exemplify "cascading errors" and explain why sampling multiple functions helps mitigate such errors?**
>
> In this context, "cascading errors" refer to where an error in the sub-function causes the parent function to also fail. For example, a badly implemented 'cosine' function will cause everything depending on it to malfunction.
>
> As discussed in W2, sampling implementations on a single function can reduce its incidental errors, so in decomposition it would also reduce overall (cascading) errors in the whole program.
>
> > **Q2: Can the selection of benchmarks really exercise the core part of solving complicated coding tasks?**
>
> Following AlphaCode and CodeLLAMA's definition of 'complex' competition-level datasets, we used xCodeEval and MATH in our evaluations. Table 1, 4, 10 show that our method achieves significant improvements on these more difficult problems.
>
> However, we fully agree with your concerns regarding complex problems. Due to the scope of our work, we haven't yet been able to test our performance on software development tasks, which often involve engineering details like changing requirements, code retrieval, and real-time debugging. Nevertheless, we believe that the idea of divide-and-conquer and consensus can be equally applied to such complex problems and represents a promising area for future research.
>
> > **Q3a: When computing the token usage, did you include tokens of unused samples when computing "functionality similarity"?**
>
> Yes, we included the token count for unused sampled functions, and we obtained token cost directly from OpenAI API's statistics. Hence the token count reported in our paper is the total token expenditure from all API calls throughout the process of solving a problem.
>
> > **Q3b: Can you also provide more statistics such as the medium token consumption in case the average number is biased by simple tasks?**
>
> Thanks for your suggestion. We've added a lot more statistics in our General Response and we'll include these data in our next revision.
>
> > **Q3c: The token consumption is much smaller than I expected according to the large prompts (and these will be called multiple times).**
>
> Thank you for pointing out the concern about token costs. This is because the OpenAI Inference API still only charges prompt tokens once per call, when we use `n=k` sampling. We further discussed token costs in our general response, and concluded that it costs only $O(kN)$ tokens, same as *Standard* sampling.
>
> > **Q4: How's the token usage compared to other baselines?**
>
> We list in Table 11 (Appendix A.6) the token usage for all other baselines. Note that some baselines are not open-sourced and the token usage details were not reported in the respective papers, so they are not included in the table.
>
> > **Q5: Is the technique single-turn or multi-turn when being implemented for evaluation?**
>
> We used single-turn chat completions. In each call we only include a common prefix (system prompt and few-shot examples) and one assistant question describing the current task. This helps reduce token costs, context lengths and prevents context contamination.

---

> > ### Comment · Reviewer_Ar6t · 2024-08-12
> > **Thanks for the reply!**
> >
> > Overall I am satisfied with the authors' responses. Weaknesses 3 and 4 should be easily addressable and I look forward to seeing more models being compared. While weaknesses 1&2 are not resolved due to their challenges, I suggest at least discussing them as limitations of the technique in the revision and calling for further research. Specifically, I feel the concern of W1 is a bit outstanding. Probably the only way to address it right now is to add program contracts via human experts (for example, contracts are added manually in the EvalPlus paper for test-case reliability) or LLMs; some weak pre-condition is also better than nothing even if they cannot be strictly verified.
> >
> > I'm happy to increase by rating to 6.

---

> > > ### Author Response · Authors · 2024-08-12
> > >
> > > We sincerely thank you for your response and your rating update.
> > >
> > > We fully agree with you that weakness 1 & 2 should be at least discussed in the limitations section, and we will add them in our next revision. We look forward to future research that investigates these weaknesses and, better yet, be accompanied with theoretical analysis.

---

### Official Review · Reviewer_LV7h · 2024-07-08

**Soundness:** 3
**Presentation:** 3
**Contribution:** 3
**Rating:** 6
**Confidence:** 4

**Summary:**

The paper presents FuncCoder, a novel coding framework designed to enhance code generation by incorporating a divide-and-conquer strategy with functional consensus. FuncCoder addresses the limitations of existing methods that struggle with complex programming tasks by recursively breaking down problems into smaller sub-functions, which are then hierarchically organized and recomposed to achieve more complex goals. The functional consensus mechanism mitigates error propagation by selecting functions that exhibit similar behaviors. Experimental results show that FuncCoder outperforms state-of-the-art methods, such as Parcel and Reflexion, on various benchmarks, including HumanEval, MBPP, and xCodeEval, and demonstrates its effectiveness across different language models like GPT-3.5, GPT-4, as well as smaller models such as StableCode3b. The framework's ability to dynamically decompose and compose functions enables superior performance in handling complex coding requirements and mathematical reasoning tasks.

**Strengths:**

+ The paper proposes FunCoder, a plan-and-solve coding LLM optimized with recursively divide-and-conquer. The design of FunCoder is generally reasonable and the methodologies are well explained and motivated.
+ Besides outperforming SOTA baselines, such as Parcel and Reflexion based on GPT-based models, it is impressive how FunCoder significantly improve the small LLMs' performance.
+ The paper performs in-depth analysis and ablation study to illustrate FunCoder's effectiveness and justify different design choices.

**Weaknesses:**

While FunCoder demonstrates impressive performance in handling well-defined programming challenges, there is a practical concern regarding the token length of the trajectory. The recursive decomposition of tasks into sub-functions inherently leads to the generation of numerous function headers, bodies, and documentations, which can significantly inflate the token count. As reported by Table-3, the average token length is ~5.5k, and I am curious about the median and the maximum tokens required to solve HumanEval problem.

This becomes concerning when dealing with more intricate and interconnected coding requirements. As each layer of decomposition adds to the overall length, the token usage can escalate rapidly, leading to inefficiencies and increased computational costs. The empirical data shows that while FunCoder performs well on self-contained coding benchmarks, the token length could become a limiting factor for more complex problems, potentially offsetting the advantages gained through the divide-and-conquer approach. This exponential growth in token length necessitates careful consideration and optimization to ensure that FunCoder remains scalable and efficient for a broader range of coding tasks.

**Questions:**

+ Why do the authors report different methods for GPT-3.5 and GPT-4 in Table-1? Aren't these approaches generalizable to all GPT-based models through API? I would suggest the authors to add back missing baselines for both models. A similar question presents in Table-4
+ In Table-4, why do the authors choose CoT or self-refine as the main baseline while no more comparing with Parcel?
+ Why do the authors choose not to disclose their code during the submission phase?

**Limitations:**

The paper discusses the limitation in Section 5.

---

> ### Author Rebuttal · Authors · 2024-08-06
>
> We thank you for your time and effort in reviewing our paper! These are very helpful suggestions and we address your concerns as following:
>
> > **W1: The recursive decomposition of tasks into sub-functions inherently leads to the generation of numerous function headers, bodies, and documentations, which can significantly inflate the token count.**
>
> Indeed, decomposing the original problem into multiple functions may increase token usage. However:
>
> - Function headers and documentation don't occupy much tokens, since they typically just have a few lines, and are way smaller than the function body that might span to dozens of lines.
> - Programs that split functions don't have to be longer than un-split programs, due to function re-use which helps reduce redundant code.
> - Our method does not force the LLM to decompose overly simple tasks into subfunctions. In fact, the standard method also generates sub-functions, and both methods have similar resulting code lengths. Our approach just regenerates function with fine-grained divide-and-conquer above that.
>
> We might also add that, we showed in Table 11 (L759) that although our token usage was not the best of all, but that it was small enough given its state-of-the-art performance (e.g. LDB took 23K tokens to get 82.9% while ours made 85.4% with 5.4K tokens).
>
> > **W2: Missing the median and the maximum tokens required to solve the HumanEval problem.**
>
> Thank you for pointing out the flaws in our initial statistical method. We reviewed Table 3, the ablation study of FunCoder on HumanEval with gpt-3.5-turbo, re-ran statistics on the original data and obtained the following results (you can find them in our General Response):
>
> | Setting | Pass@1 | min tks | max tks | mean tks | med tks |
> | --- | --- | --- | ---- | --- | --- |
> | Standard | 68.3 | 648 | 1477 | 886.7 | 861.5 |
> | One-pass | 72.6 (+4.6) | 826 | 3489 | 1233.7 | 1132.0 |
> | Two-pass | 78.7 (+10.4) | 2197 | 8406 | 3343.2 | 3078.0 |
> | Two-pass + ST@11 | 80.5 (+12.2) | 2967 | 13758 | 5408.3 | 5184.0 |
> | FunCoder@5 | 83.5 (+15.2) | 2455 | 9432 | 4040.9 | 3800.0 |
> | **FunCoder@11** | 85.4 (+17.1) | 3015 | 13850 | 5402.0 | 5166.5 |
>
> > **W3: When tackling complex problems, the token consumption grows exponentially.**
>
> We did not provide a token complexity analysis in the original paper, but our method actually costs only O(N) tokens and it linearly scaled to the size of the generated program. We discussed this proof in detail in general response, and this will be added to the next revision of this paper.
>
> > **Q1: Why baselines are different on GPT-3.5 and GPT-4?**
>
> Some results are directly obtained (and cited) from the original papers, and we've marked them with an underline in Table 1. Some methods only reported results for GPT-4, and upon careful examination, results for other models were not mentioned in these papers. In Appendix A.2, we provided details for which results were derived from the original papers, and specified whichever methods were specifically tailored for code generation or mathematical reasoning tasks.
>
> Some methods compared are not fully open-source or lack detailed instructions. Since we weren't able to reproduce them consistently, we had to cite them verbatim instead. We'd like to add more experiments in the next revision and get more updates on this topic.
>
> > **Q2: In Table-4, why do the authors choose CoT or self-refine as the main baseline while no more comparing with Parsel?**
>
> Parsel did not provide prompts or code for tasks beyond code generation with HumanEval, and it is not designed specifically for mathematical reasoning. Therefore, we did not compare Parsel on the math tasks. Instead, we introduced baselines that were specifically designed for mathematical reasoning, which are more commonly used in that area, such as CoT, self-refine, and CR.
>
> - Standard and CoT serve as text-based baselines for mathematical reasoning, compared with other methods that solve math problems through code generation.
> - Self-Refine uses runtime feedback to fix code, serving as a baseline for comparison with our sampling-based approach.
> - CR (Cumulative Reasoning) is another baseline for step-by-step problem solving and was the previous SOTA method for the MATH dataset. Unlike our approach, which employs a divide-and-conquer strategy, CR uses a bottom-up technique that progressively derives the final answer.
>
> Details about these methods are provided in Appendix A.2 (L627).
>
> > **Q3: Why do the authors choose not to disclose their code during the submission phase?**
>
> We are still organizing our codebase during the paper submission phase. However, we are confident that we can prepare a minimal working code and accompanying bash scripts for replicating the experiments within the next few months. Meanwhile, as the principles of our method are relatively straightforward, to ensure reproducibility of our work, we have provided complete prompts verbatim so that just calling an API with these prompts would suffice. These prompts should work with most methods and models unless specifically noted.

---

> > ### Comment · Reviewer_LV7h · 2024-08-12
> >
> > Thanks for the authors response. It mostly makes sense, though I have concerns why the minimum working code for replication requires **months** to release while the authors acknowledge the methods are relatively straightforward. I would like to see some brief explanations of what are the most time-consuming part for code release and the main difficulty of implementing the current agentic system.

---

> > > ### Author Response · Authors · 2024-08-13
> > >
> > > We thank you for your response and again apologize for causing confusion regarding our code.
> > >
> > > To put it honestly, we were quite busy with other things on our hands, so didn't have enough time to polish our code. But we reassure you that the method **and the implementation** are still straightforward, and that replicating this work from scratch needn't two months time (far less than that).
> > >
> > > As is also discussed in the *Limitations and Social Impact* section that directly running code generated by LLMs can be risky. Although our current version of Code Interpreter Sandbox works and could prevent the machine from many known attacks, we are actively looking for more promising means of protection, lest our code causes problems when it goes public.
> > >
> > > Comments and suggestions on how the code should be open-sourced are also super welcomed.

---

### Official Review · Reviewer_8URc · 2024-07-10

**Soundness:** 3
**Presentation:** 3
**Contribution:** 3
**Rating:** 7
**Confidence:** 3

**Summary:**

The author applies divide-and-conquer methods to code generation problems with Large Language Models (LLMs). A problem is divided into subproblems recursively; that is, the function that solves the problem is generated in a top-down manner. Given the parent function, the LLM is prompted to implement it using child functions, which are yet to be implemented. Then, it recursively implements the child functions. There are two key components that make this method work well:

1. A two-pass generation approach: When generating a function f, the first pass generates the function/plans without child functions being implemented yet. The second pass occurs when the children functions are implemented, conditioning on the children functions and regenerating the parent functions again. This is to overcome potential problems that may arise when parent functions are generated before child functions are concretely implemented.
2. Consensus via multiple sampling: Each time a function is regenerated in the second pass, multiple programs are sampled, and a consensus of the samples is used to generate the final program. The consensus is determined by selecting the program that exhibits the most similar input-output behavior to other functions.

Experimental results show it improves performance effectively on HumanEval, MBPP, xCodeEval, and MATH benchmarks, and remarkable 94.5% results on HumanEval with GPT4.

**Strengths:**

* The method is conceptually very clean and effective empirically, as shown on various benchmarks including three code generation tasks and math-solving tasks.
* Extensive experiments and ablation studies are conducted to show the effectiveness of the method.
* The results also show that the method is effective not just on large closed-source models, but also on smaller open models such as LLaMA 3 8B and StableCode 3B. This contributes to the reproducibility of the method.

**Weaknesses:**

* The idea of decomposing the problem into subproblems and solving them recursively with an LLM is not entirely new.
* The consensus method, which aims to select top programs from a set of samples using similarity, is different but similar to the AlphaCode approach. It would be interesting to see a comparison of selecting the top programs using majority voting on the input/output as in AlphaCode instead of calculating the similarity.

**Questions:**

* On average, how many decomposition levels are used when solving the code generation tasks?
* Are the Standard and CodeT prompts used in the experiments the ones shown in Appendix C.1? Is this type of decomposition prompting also better for the Standard and CodeT methods?
* How does the consensus method compare to other self-consistency or AlphaCode-like majority clustering methods?
* In the abstract and line 167, it is said that GPT4 performance on HumanEval is about 97% but I can't find that number in Table 1. Is it mentioned else where?

**Limitations:**

Yes, one limitation that it is not suitable for software engineering/ open coding problems is discussed.

---

> ### Author Rebuttal · Authors · 2024-08-06
>
> We sincerely thank you for your time and effort in reviewing our paper! We find your suggestions very helpful and we address your questions as follows:
>
> > **W1: The idea of decomposing the problem into subproblems and solving them recursively with an LLM is not entirely new.**
>
> As is discussed in *4. Related work*, there is indeed previous work that decomposes problems into subproblems and explores them through search. But many of those focus on mathematical reasoning or multi-hop question answering, and can have difficulties applying to code generation.
>
> Specific to code generation, we innovatively leveraged the inherent advantages of functions, using function signatures to identify sub-problems. We further relate sub-problems via function declaration and invocation, thus decomposing complex tasks in a more natural way. We believe that this form of task decomposition can have a more straightforward advantage in code generation and can better elicit the potential of code models.
>
> > **W2: Comparison between consensus and clustering.**
>
> We find your suggestion to be very constructive. We have made a comparison between these methods on `gpt-3.5-turbo` and have observed a considerable advantage in functional consensus:
>
> | Method \ Pass@1 | HumanEval | MBPP | xCodeEval | MATH |
> | --- | --- | --- | --- | --- |
> | Standard | 68.3 | 72.0 | 20.2 | 34.6 |
> | CodeT | 81.1 | 76.0 | 23.2 | n/a |
> | Clustering | 75.0 | 74.0 | 22.0 | 35.0 |
> | FunCoder (ours) | 85.4 | 78.5 | 31.4 | 54.0 |
>
> *Clustering* from AlphaCode strictly classifies programs into mutually independent groups by exactly identical outputs. Our method measures output similarity based on many input cases, which is a more permissive approach. Since inputs to a program may have edge cases, treating two programs distinctly just because one output was different, without considering how similar they are, is sub-optimal. For example:
>
> - Consider finding all square roots of a float. Generate 10 functions.
> - 4 results know that a positive number has two square roots. 4 results know that a negative number has imaginary square roots. Only 2 results considered both points.
> - Notice that the model gets 60% right for either point independently, but only 20% when the points are put together.
> - Clustering would put programs in three categories [4, 4, 2] based on output, and end up choosing who only gets one point right. But with our function similarity, the one who gets both points right that is on-more-aspects similar will be favored.
>
> Getting benefits on *Clustering* often requires more programs (even up to 1M in AlphaCode paper). Our method does well with just 11 sampled programs. Consensus is also exceptionally good at pass@1 while AlphaCode focused on pass@10 instead.
>
> > **Q1: On average, how many decomposition levels are used when solving the code generation tasks?**
>
> We analyzed the depth of functions from the results and show them in this table:
>
> | | HumanEval | MBPP | xCodeEval | MATH |
> | --- | --- | --- | --- | --- |
> | (avg) Depth | 1.19 | 1.06 | 1.45 | 1.34 |
>
> Note that since we rely on the intrinsic capabilities of the model to determine whether or not to decompose functions during generation, we cannot forcibly control the depth of decomposition. The LLM tends to generate shallow code for simple problems and vice versa, which is why the average depth is just slightly above 1.0. But this method is quite effective on complex problems exhibiting deeper function depths.
>
> > **Q2a: Are the Standard and CodeT prompts used in the experiments the ones shown in Appendix C.1?**
>
> Yes, both Standard and CodeT used the prompts from Appendix C.1. We will clarify this point in a later revision and explicitly specify the prompts used for each method on HumanEval.
>
> > **Q2b: Is this type of decomposition prompting also better for the Standard and CodeT methods?**
>
> To ensure the fairness between the decompose prompt and the standard prompt, we used the same example in all prompts. So the *Standard* method also breaks down the original problem into multiple functions, but it generates all the functions all at once, whereas our *decomposition* prompting allows for recursive sub-function generation. However, it's worth noting that the *decomposition* prompt always requires iterative-decomposition, since it only produces sub-function stubs and would otherwise produce incomplete programs.
>
> In Table 3, we conducted an ablation study on the results of decomposition. "One-pass" refers to just *Divide* (recursively decomposing) functions above *Standard*, and "Two-pass" refers to having both *Divide* and *Conquer* but without functional consistency applied. The results indicate that applying just recursive decomposition will still yield a considerable improvement.
>
> > **Q3: How does the consensus method compare to other self-consistency or AlphaCode-like majority clustering methods?**
>
> Thanks for your suggestion. Clustering in AlphaCode can be viewed as self-consistency over programs in code generation. So we re-ran the experiments with majority clustering mentioned in AlphaCode, and you can see the results in our response to Weakness 2.
>
> > **Q4: In the abstract and line 167, it is said that GPT4 performance on HumanEval is about 97% but I can't find that number in Table 1. Is it mentioned elsewhere?**
>
> We apologize for any confusion caused by our wording. What we intended to convey is that on the HumanEval dataset, StableCode-3b with FunCoder (81.0% pass@1) achieved 97.7% of the performance of GPT-4 with Standard (82.9% pass@1). This statement in the abstract aims to highlight that FunCoder can significantly enhance the code generation performance of small open-source models, bringing them close to that of advanced models, which to our belief was a very interesting finding.
>
> A similar point is also discussed in section 3.1.3 (L163-167). We apologize again for any misunderstanding caused by any ambiguity in our paper, and we shall clarify this in a future revision.

---

### Author Rebuttal · Authors · 2024-08-07

We thank all the reviewers for taking the time and effort in reviewing our paper, and we find these comments very constructive and inspiring. Hereby we address some of the most common concerns and questions, adding additional experiments and analyses as-appropriate. We hope that this information will provide you with more insights into our methods, and we're certainly welcome to further discussions on these topics.

## 1. Results on More Models

We thank reviewer Ar6t for the suggestion regarding model diversity. We've been paying close attention to new cutting-edge models and have supplemented our experiments accordingly.

| Model | Method \ Pass@1 | HumanEval | MBPP | xCodeEval | MATH |
| --- | --- | --- | --- | --- | --- |
| GPT-4o mini | Standard | 87.2 | 76.0 | 35.4 | 51.4 |
| | FunCoder | 91.5 | 77.5 | 39.8 | 52.6 |
| Codestral 22B | Standard | 79.3 | 68.5 | 11.4 | 31.4 |
| | FunCoder | 89.0 | 74.5 | 22.0 | 36.8 |
| StarCoder2 15B | Standard | 59.8 | 64.5 | 7.2 | 21.0 |
| | FunCoder | 78.7 | 70.0 | 11.6 | 28.8 |

## 2. Analysis of token cost

### 2.1 Example

We use the example from Figure 1, where the final program consists of 5 functions A[B[D,E],C], and A serves as the entry to the program. We denote $N=A+B+C+D+E$ as the token complexity of this task. The order in which calls are executed is put in a pair of parentheses before each line in the following examples.

**Standard:** Call once only. Completes the given function.
```
(1) a -> ABCDE
input tokens = a
output tokens = A+B+C+D+E
overall = O(N)
```

**FunCoder/Divide:** In each step of the *Divide* stage the to-be-implemented function will serve as the context. The function will be implemented and sub-function stubs will be declared.
```
(1) a -> Abc
(2) b -> Bde
(3) d -> D
(4) e -> E
(6) c -> C
input tokens = a+b+c+d+e
output tokens = A+b+B+c+C+d+D+e+E < 2N
overall = O(N)
```

**FunCoder/Conquer:** Here the context includes the current function's definition and finalized implementations of sub-functions. The output is the re-implemented current function.
```
(5) bDE -> B
(7) aBC -> A
input tokens = a+b+B+C+D+E < 2N
output tokens = A+B
```

**FunCoder/Consensus:** When sampling (`n=k`) is enabled in the bottom-up process, the *Conquer* stage automatically includes 'consensus'. Here input tokens are still counted once, and output tokens are counted k-times.
```
(3) d -> kD
(4) e -> kE
(5) bDE -> kB
(6) c -> kC
(7) aBC -> kA
input tokens = a+b+B+c+C+d+D+e+E < 2N
output tokens = kA+kB+kC+kD+kE = kN
overall = O(kN)
```

**Conclusion:** The token cost involved in FunCoder and Standard are both worst-case $O(N)$ which linearly scales to the number of final output tokens (i.e. problem's inherent complexity). Even if sampling is applied its complexity $O(kN)$ would still be on par with other methods (e.g. self-consistency, CodeT) where sampling is also enabled.

### 2.2 Complexity of token-cost

We explain that the worst-case token cost of our method is $O(kN)$. For the sake of simplicity, consider the case without output sampling:

- Suppose that the program will have $N$ tokens.
- We ignore the tokens involved with prompting since they are generally proportional to the number of LLM calls.
- The naive 'Standard' method should generate exactly $N$ tokens.
-  FunCoder goes through the *Divide* stage and the *Conquer* stage for each of the functions. Without loss of generality,
  - Based on the current function, *Divide* generates an implementation of itself and stubs for sub-functions. In this stage, each function would appear at most once in input and twice in output. All *Divide* stages consume no more than $3N$ tokens.
  - Recursively generate all sub-functions, and
  - *Conquer* regenerates the parent function based on its stub and all finalized sub-functions. Here each function will appear at most twice in input and exactly once in output. All *Conquer* stages shall consume at most $3N$ tokens.

So FunCoder requires no more than $6N$ tokens in input-output, making its token consumption $O(N)$ even at worst-case.

We further argue that even if output sampling is enabled (`n=k`), this complexity will still be a linear $O(kN)$. This shows that our token complexity is consistent with other sampling-enabled methods like Standard + CodeT.

### 2.3 Detailed result of token cost distribution

We thank reviewer LV7h and s7Dq for their suggestions on more comprehensive statistics regarding token consumption. We have added minimum-maximum values, median and distribution to the statistics, included in the table below. This further supports the point that our method merely applies a constant factor to the token cost.

| Setting | Pass@1 | min tks | max tks | mean tks | med tks |
| --- | --- | --- | ---- | --- | --- |
| Standard | 68.3 | 648 | 1477 | 886.7 | 861.5 |
| One-pass | 72.6 (+4.6) | 826 | 3489 | 1233.7 | 1132.0 |
| Two-pass | 78.7 (+10.4) | 2197 | 8406 | 3343.2 | 3078.0 |
| Two-pass + ST@11 | 80.5 (+12.2) | 2967 | 13758 | 5408.3 | 5184.0 |
| FunCoder@5 | 83.5 (+15.2) | 2455 | 9432 | 4040.9 | 3800.0 |
| **FunCoder@11** | 85.4 (+17.1) | 3015 | 13850 | 5402.0 | 5166.5 |

These results will be included in the next revision of our paper.

## 3. Exemplify why Consensus Works

We thank reviewers 8URc and Ar6t for pointing this out and would like to elaborate further:

- Consider finding all square roots of a float. Generate 10 functions.
- 4 results know that a positive number has two square roots. 4 results know that a negative number has imaginary square roots. Only 2 results considered both points.
- Notice that the model gets 60% right for either point independently, but only 20% when the points are put together.
- With our function similarity, the solution which is on-more-aspects similar (and happens to be correct) will be favored.

Through this example, we illustrate that Functional Consensus has the potential to identify the correct samples even at their minority, outperforming other methods such as self-consistency or clustering.

---

### Decision · Program_Chairs · 2024-09-25

**Decision:**

Accept (oral)

**Comment:**

This paper introduces FunCoder, an elegant divide-and-conquer strategy for code generation which uses an LLM to recursively decompose a complex programming task into a tree of functions calling helper functions. Because the units of decomposition are individual code functions, FunCoder also uses the notion of functional consensus to select good helper function implementations before completing the implementation of the parent function. These techniques lead to significant improvements in multiple code generation and math reasoning benchmarks.

I recommend to **accept** this paper because, in my opinion, this is a great paper with many strengths:
* The topic of LLM-based code generation for complex tasks is clearly important.
* Reviewers agreed that FunCoder is "conceptually very clean", "novel and beautiful", and "well explained and motivated".
* Reviewers also praised the "extensive experiments" and their "striking" results: "Besides outperforming SOTA baselines, ... it is impressive how FunCoder significantly improve the small LLMs' performance."
* Beyond the impressive end-to-end results, the authors dive deeper to better understand the method and explain its benefits compared to prior approaches. In this way, the paper "justif[ies] different design choices" and "provides a number of interesting insights beyond the base experimental results".
* The paper's writing is "commendably clear". In fact, I think a reader can understand the main divide-and-conquer idea from only Figure 1 and Figure 2.
* The authors provided detailed rebuttals containing clarifications, further analyses, and new experimental results. These rebuttals strengthened the paper's conclusions and improved the reviewers' opinions of the work.
* After discussions, all reviewers are supportive of accepting this paper.

Most weaknesses raised by reviewers were addressed by the rebuttals, but a minor weakness remains: the method is not evaluated on realistic complex programming tasks, despite being designed for complex tasks. A full experiment is a lot to ask for given the paper's broad scope already, but I think it would be very enlightening to include a small case study on 1-3 carefully-chosen realistic complex tasks (chosen based on the needs/strengths of FunCoder and simplifying assumptions like being contained in one file, not cherrypicked post-hoc based on results).

I also think this paper should be highlighted (more than a poster) because its ideas are intuitive, elegant, and broadly applicable. Decomposition in program synthesis is not a new subject, but the divide-and-conquer strategy proposed in this paper is a novel and conceptually simple approach that seems generally applicable to more languages and programming domains than explored by the paper. The concept of functional consensus is also a refreshing alternative to clustering of function behavior (majority voting) or self-testing (like CodeT). I anticipate that the core ideas in this paper could be easily communicated to and appreciated by a broader audience without requiring much prerequisite knowledge, another reason for this paper being well-suited for an oral presentation.

---

I also want to share some thoughts with the authors, in case it helps improve a future version of the paper. Your rebuttal includes an example illustrating why functional consensus works. Here is an additional explanation which may be more intuitive.

Functional consensus reminds me of "special-case similarity" from the FrAngel paper (Section 4.1 of https://arxiv.org/pdf/1811.05175). In short, this is a common property where a fully-correct program is similar to other (simpler) programs that only solve a subset of the desired functionality (a "special case"). The FrAngel paper found success in program synthesis search by remembering the simplest program generated so far which solves each different subset of tests, and biasing generation of future programs toward the set of remembered programs. This works if enough remembered programs contain sufficient similarity to a fully-correct solution to increase the chances of progress.

Bringing this back to functional consensus, a possible explanation for its effectiveness is as follows. Consider a correct solution program which has multiple "special-case programs" which are partially correct in different ways, for example:

* a program that behaves correctly in the general case but misses an edge case
* a program that handles the edge case correctly but messes up the general case
* a buggy program that behaves correctly for all tests that don't trigger the bug
* a program which replaces an if/else block with just the body of the else, which still behaves correctly for the subset of tests where the condition evaluates to false
* a program with a mix of the above properties

If we had a pool of programs containing the fully-correct program along with multiple special-case programs such as those listed above, then we would expect:

* the fully-correct program would be similar (in behavior on tests) to every other special-case program
* each special-case program would be similar to other special-case programs only to the extent that the special cases themselves overlap, which is less than the amount of overlap with the fully-correct program

So, intuitively we would expect that the fully-correct program is selected by functional consensus.

It may also be interesting to weight each test by how much it's able to distinguish between different sampled programs, considering the other tests as well. For example, the first test that exercises some edge case is really helpful in identifying programs that handle that edge case correctly. But further tests for the same edge case may not provide any additional "distinguishing power". Should an edge case have more impact on functional similarity simply because it has more tests? A correct program must handle the edge case correctly, no matter how many tests illustrate that edge case. But, perhaps failing a rare edge case is more acceptable than failing a common edge case. The right decision might depend on how exactly the tests are generated.